# Spatiotemporal control of actomyosin contractility by MRCKβ signaling drives phagocytosis

Ceniz Zihni[1], Anastasios Georgiadis[1,2], Conor M. Ramsden[1], Elena Sanchez-Heras[1], Alexis J. Haas[1], Britta Nommiste[1], Olha Semenyuk[1,2], James W.B. Bainbridge[1,2,3], Peter J. Coffey[1], Alexander J. Smith[2], Robin R. Ali[1,2,3], Maria S. Balda[1], and Karl Matter[1]

**Phagocytosis requires actin dynamics, but whether actomyosin contractility plays a role in this morphodynamic process is unclear. Here, we show that in the retinal pigment epithelium (RPE), particle binding to Mer Tyrosine Kinase (MerTK), a widely expressed phagocytic receptor, stimulates phosphorylation of the Cdc42 GEF Dbl3, triggering activation of MRCKβ/myosin-II and its coeffector N-WASP, membrane deformation, and cup formation. Continued MRCKβ/myosin-II activity then drives recruitment of a mechanosensing bridge, enabling cytoskeletal force transmission, cup closure, and particle internalization. In vivo, MRCKβ is essential for RPE phagocytosis and retinal integrity. MerTK-independent activation of MRCKβ signaling by a phosphomimetic Dbl3 mutant rescues phagocytosis in retinitis pigmentosa RPE cells lacking functional MerTK. MRCKβ is also required for efficient particle translocation from the cortex into the cell body in Fc receptor–mediated phagocytosis. Thus, conserved MRCKβ signaling at the cortex controls spatiotemporal regulation of actomyosin contractility to guide distinct phases of phagocytosis in the RPE and represents the principle phagocytic effector pathway downstream of MerTK.**

## Introduction

Phagocytosis is an ancient biological process that is essential for the functions of a variety of different cell types ranging from macrophages and microglia to specialized epithelia such as the retinal pigment epithelium (RPE; Chaitin and Hall, 1983; Inana et al., 2018; Sparrow et al., 2010). Phagocytosis is triggered by cell surface receptors binding to a target particle, often in concert with coreceptors, stimulating sequential morphological changes leading to particle internalization (Freeman and Grinstein, 2014; Jaumouille et al., 2019; Swanson, 2008). Some phagocytes, such as macrophages, express a repertoire of specialized receptors with different specificities, such as Fc receptor (FcR), to phagocytose different types of particles. Other phagocytic receptors are widely expressed and are found in most phagocytic cells. This includes the TAM receptor Mer Tyrosine Kinase (MerTK) that regulates phagocytosis by phagocytes of the immune system as well as microglia, Schwann cells, and the RPE. Hence, deregulation of MerTK-regulated phagocytosis has been linked to diseases, ranging from cancer and rheumatoid arthritis to neurodegenerative conditions and blindness (Brosius Lutz et al., 2017; Myers et al., 2019; Haukedal and Freude, 2019; D'Cruz et al., 2000). However, the downstream signaling mechanism by which MerTK drives membrane morphodynamics to internalize particles is unknown.

In the eye, phagocytosis by the RPE is essential for clearing shed photoreceptor outer segment (POS) debris as part of a diurnal renewal process of photoreceptor membranes damaged by light (Young, 1967). Particle binding to the RPE membrane requires αvβ5 integrin (Finnemann et al., 1997; Finnemann and Rodriguez-Boulan, 1999; Lin and Clegg, 1998; Miceli et al., 1997) and MerTK, with the integrin being required for binding but not internalization of POS (Chaitin and Hall, 1983; D'Cruz et al., 2000; Edwards and Szamier, 1977; Mullen and LaVail, 1976; Ramsden et al., 2017). Internalization is stimulated by MerTK, a step essential for phagocytosis; hence, mutations inactivating MerTK lead to retinitis pigmentosa, an inherited form of retinal degeneration (Sparrow et al., 2010). Defects in phagocytosis also contribute to age-related diseases in the eye due to deregulation of mechanisms that are poorly understood (Inana et al., 2018; Sparrow et al., 2010).

Despite the diversity of phagocytic receptors, cell types, and target particles' properties, particle engulfment always requires a dynamic actin cytoskeleton. Engulfment is generally initiated by induction of membrane protrusions, such as the pseudopod-like structures induced by Cdc42 in FcR-mediated phagocytosis in macrophages, that then mature into a phagocytic cup wrapping the particle, which is followed by cup closure and internalization (Barger et al., 2020). The RPE membrane generates

[1]UCL Institute of Ophthalmology, University College London, London, UK;   [2]Gene and Cell Therapy Group, UCL Institute of Ophthalmology, University College London, London, UK;   [3]National Institute for Health and Care Research Biomedical Research Centre at Moorfields Eye Hospital National Health Service Foundation Trust, London, UK.

Correspondence to Ceniz Zihni: c.zihni@ucl.ac.uk;   Maria S. Balda: m.balda@ucl.ac.uk;   Karl Matter: k.matter@ucl.ac.uk.

pseudopod-like protrusions (Jiang et al., 2015), topologically similar to protrusions generated in FcR-mediated Cdc42-driven macrophage phagocytosis.

Although actomyosin contractility is central to a broad range of membrane remodeling processes, its role in phagocytosis is not clear. In macrophages, the role of myosin is controversial in both fragment crystallizable region– and complement receptor–mediated phagocytosis, and recent studies suggest that actomyosin contractility may not be required for cup morphogenesis and internalization (Barger et al., 2020; Jaumouille et al., 2019; Rotty et al., 2017). Moreover, the molecular mechanisms that spatially and temporally regulate actomyosin activity and potentially couple it to distinct morphological transformations to control distinct phases of particle internalization have not been defined in any phagocytic cell type.

Here, we identify the engulfment pathway downstream of MerTK receptor activation and identify an important role for actomyosin activity in RPE phagocytosis. MerTK activates the Cdc42 guanine nucleotide exchange factor (GEF) Dbl3 through Src-dependent phosphorylation. Locally activated Cdc42 then stimulates two effectors, N-Wiskott–Aldrich syndrome protein (N-WASP) and MRCKβ. N-WASP stimulates actin-based pseudopod-like protrusions and MRCKβ activates myosin-II to limit actin polymerization and control deformation of pseudopods into cups. MRCKβ-driven contractility also stimulates assembly of a mechanosensory bridge consisting of activated FAK, talin, and vinculin; and integrin clustering to guide membrane wrapping around particles. Finally, MRCKβ drives completion of cup closure and particle internalization. The functional importance of the discovered mechanism is further supported by its requirement to maintain retinal integrity in vivo. Direct activation by expression of a phosphomimetic Dbl3 mutant that circumvents the requirement for MerTK rescues phagocytosis in induced pluripotent stem cell (iPSC)–derived retinitis pigmentosa RPE cells, indicating that the principal function of MerTK in phagocytosis is activation of the MRCKβ pathway. MRCKβ signaling is also required for efficient translocation of internalized yeast particles from the cell cortex to cell body during FcR-mediated phagocytosis in macrophages, indicating a conserved yet diverse role for MRCKβ in regulating the actomyosin cortex during phagocytosis in distinct phagocytes.

## Results

### Conserved cortical Cdc42 signaling drives phagocytosis in the RPE

Phagocytosis in the RPE involves the generation of pseudopod-like protrusions with a morphology similar to those in macrophages induced by particle binding to FcRs (Jiang et al., 2015). Macrophages require Cdc42 signaling to initiate phagocytosis (Barger et al., 2020). The apical membrane of epithelia is enriched in evolutionarily conserved Cdc42 signaling components; however, their postmitotic functions once cells have polarized are not known. Therefore, we tested whether the apical Cdc42 GEF Dbl3 and its downstream effector MRCKβ, a myosin-

II activator, are involved in phagocytosis (Kumfer et al., 2010; Marston et al., 2016; Zihni, 2021; Zihni et al., 2016; Zihni et al., 2014; Zihni and Terry, 2015; Zihni et al., 2017).

Depletion of Dbl3 in confluent polarized porcine primary RPE cells using two distinct siRNAs resulted in inhibition of POS internalization and reduced phagocytic efficiency (Fig. 1, a–c, and Fig. S1, a–c). POS binding (the total sum of POS-FITC attached to the cell membrane and internalized) to the RPE was not affected, indicating normal receptor expression (Fig. 1 b). Depletion of Cdc42 also resulted in inhibition of POS internalization (Fig. 1, d and e, and Fig. S1, d and e). Staining with anti-Cdc42-GTP antibodies (Elbediwy et al., 2012) revealed that POS binding stimulated local Cdc42 activation at attachment sites following 20-min incubations, which was strongly attenuated by knockdown of Dbl3, indicating that Dbl3 activates Cdc42 at the onset of POS/membrane binding (Fig. 1 f and Fig. S1 f).

To examine how Cdc42 signaling initiates particle engulfment, we employed ARPE-19 cells, which normally phagocytose very slowly, as a gain of function model (Ablonczy et al., 2011). Exposure of ARPE-19 cells to POS-FITC for 1 h followed by a 1 h chase resulted in efficient binding to the apical membrane, but protrusion induction and particle internalization were inefficient (Fig. 1, g–i). In contrast, exogenous expression of Dbl3-myc led to prominent POS-dependent nascent protrusions, phagocytic cups, and POS internalization (Fig. 1, h and i, and Fig. S1 g). Dbl3-myc localized along these nascent protrusions up to the tips (Fig. 1 i, and Fig. S1 g; red arrowheads). Knockdown of Cdc42 and MRCKβ in Dbl3-myc expressing ARPE-19 cells confirmed the conserved role of Dbl3 as the apical GEF activating Cdc42/MRCKβ signaling (Fig. 1 h and Fig. S1 h).

N-WASP plays an important role in actin nucleation and polymerization by activating ARP2/3 in response to binding active Cdc42 (Carvalho et al., 2013; Miki et al., 1998). N-WASP depletion inhibited POS phagocytosis, but not binding, in cells expressing Dbl3-myc (Fig. 1, g and h, and Fig. S1 h). Examination after 30 min of POS incubation revealed enrichment of both Dbl3-myc and F-actin at the apical cortex in proximity to POS membrane binding sites, which was inhibited by N-WASP siRNA (Fig. 1 j and Fig. S1, i–k). Therefore, RPE phagocytosis requires POS-induced Cdc42 signaling, stimulating two effectors, the myosin-II activator MRCKβ and the regulator of actin polymerization N-WASP.

Analysis of Cdc42 signaling components at nascent protrusions revealed enrichment of Dbl3, Cdc42-GTP, MRCKβ, and N-WASP, when compared to non–POS-bound membrane areas (Fig. 1, k–m, and Fig. S2, a–d). The Cdc42-signaling components continued to localize at mature phagocytic cups (Fig. 1, k and l). Time course analysis in primary RPE cells of integrin αvβ5 localization, the POS receptor, revealed enriched labeling at POS attachment sites at the onset of POS/RPE contact (10 min; Fig. 2 a and Fig. S2 e). Integrin αvβ5 labeling intensity rapidly increased at attachment sites, peaking at 20 min (Fig. 2 a and Fig. S2 e), correlating with an increase in nascent protrusions (Fig. 2, a–c, and Fig. S2, a–d). At 30 min, there was an increase in protrusions that adopted a cylindrical cup morphology, encircling POS (Fig. 2, a–c, and Fig. S2 e, bottom panels). Therefore, POS contact with the membrane induced αvβ5 integrin clustering at adhesion sites,

Figure 1. **Apical Cdc42 signaling drives cup formation and engulfment in the RPE. (a–e)** siRNA-mediated knockdown of Dbl3 or Cdc42 in primary porcine RPE results in inhibition of phagocytosis. Note, in cells treated with Dbl3 siRNA, apical Dbl3 expression is lost. White arrowheads highlight apical F-actin cortex. Green arrowheads highlight the position of internalized or external bound POS. **(f)** siRNA-mediated knockdown of Dbl3 in primary porcine RPE inhibits Cdc42 activation. **(g–i)** Exogenous expression of Dbl3-myc in ARPE-19 cells stimulates increased POS-induced pseudopod and cup assembly, and phagocytosis via Cdc42, MRCKβ, and N-WASP signaling. White arrowheads highlight the apical F-actin cortex, green arrowheads highlight the position of POS, and red arrowheads Dbl3-myc. **(j)** N-WASP depletion in Dbl3-myc expressing ARPE-19 cells followed by 30 min of POS stimulation inhibits F-actin assembly at the

vicinity of POS adhesion. White arrowheads indicate POS and green arrowheads F-actin. **(k–m)** POS-membrane contact rapidly induces membrane protrusions enriched in the Cdc42 GEF Dbl3, active Cdc42-GTP, and the Cdc42 effectors MRCKβ and N-WASP. Quantifications: means ±1SD of $n$ = 3 independent experiments; indicated are the total number of cells analyzed and P values derived from $t$ tests.

a common feature of phagocytic receptors (Sobota et al., 2005), followed by formation of membrane protrusions and membrane remodeling into cups within 30 min (Fig. 2 c).

During induction of nascent protrusions, colocalization of integrin αvβ5 with POS was almost complete (Fig. 2, d and e, and Fig. S2 f), whereas at cups colocalization was restricted to the POS-cup interface (Fig. 2, d and e, and Fig. S2 g). The enrichment of Dbl3-activated Cdc42 signaling components followed the same pattern with almost complete colocalization at nascent protrusions and was restricted to contacts at cups (Fig. 2, f–j). Analysis along the z-axis of POS confirmed that activated Cdc42 localized along the complete length and width with POS (Fig. 2, k and l). These results indicate that POS adhesion stimulates recruitment and activation of Cdc42 signaling components that drive cup assembly and maturation.

## MRCKβ controls actomyosin dynamics to guide particle wrapping in the RPE

MRCKβ phosphorylates myosin light chain (MLC) and thereby activates myosin-II. Analysis of MLC phosphorylation and F-actin staining revealed maximal phosphorylation after 30 min when phagocytic cups had formed, and POS was centered within the cups (Fig. 3, a–f). Both MLC phosphorylation and F-actin enrichment steadily decreased with internalization, which proceeded in two stages: a first one in which POS appeared embedded within the plasma membrane that peaked at 60 min and was associated with low but significant MLC phosphorylation, and a second one in which POS was in the cytoplasm and no longer positive for MLC phosphorylation. MRCK inhibition with BDP5290, which has previously been demonstrated to display greater specificity toward MRCK than related Rho-associated kinases (ROCK; Unbekandt et al., 2014), abrogated MLC phosphorylation at forming cups, and prevented the POS-induced increase in total phosphorylated MLC levels (Fig. 3, g–l, and Fig. S2, h and i). siRNA-mediated knockdown of MRCKβ also resulted in inhibition of POS-induced total phosphorylated MLC levels (Fig. 3 m). However, pseudopods still formed but failed to remodel and wrap around POS. No defect in POS binding was observed (Fig. 3 h and Fig. S2, h and i). Therefore, MRCK activity is not required for pseudopod induction but for remodeling of protrusions to form cups, particle wrapping, and internalization.

We next asked whether myosin-II activity is indeed required to control cup morphogenesis. Reversible myosin-II inhibition using blebbistatin (Kovacs et al., 2004) resulted in disorganized pseudopods and loss of efficient POS centering after 30 min (Fig. 3, n–p). Washout of blebbistatin rapidly restored normal cup organization and POS centering in the absence, but not in the presence of the MRCK inhibitor, indicating that MRCK was required upstream of myosin-II activity for phagocytic cup formation.

We next tested whether myosin-II activity was required for internalization. Cells were incubated with POS for 1 h followed by a 2 h chase to ensure internalization. In control cells, POS was efficiently internalized (Fig. 3, q–v, and Fig. S2, j and k). The addition of blebbistatin resulted in F-actin–rich disorganized pseudopods with POS accumulating around these structures and reduced internalization (Fig. 3, q–v). Accordingly, blebbistatin strongly reduced the efficiency of POS-FITC internalization (Fig. S2, j and k). Thus, myosin-II activity was required for POS internalization.

We next investigated whether MRCK activity was required using the same protocol but combined with either MRCK inhibitor BDP5290 or siRNA-mediated MRCKβ knockdown (Unbekandt et al., 2014; Zihni et al., 2017). Analysis of confocal z-sections and transmission electron microscopy (TEM) micrographs of cells incubated with Gold-labeled POS revealed that both methods of MRCK inactivation inhibited POS internalization (Fig. 4, a–d, h, j, and n; and see Fig. S3, a–c for MRCKβ siRNA deconvolution). As in shorter time courses, MRCKβ was not required for POS attachment to the membrane (Fig. 4, e and k). Thus, MRCK regulates actomyosin dynamics during cup morphogenesis and maturation to facilitate POS internalization.

MRCK inactivation resulted in an irregular apical membrane at POS-binding sites with elongated F-actin–positive protrusions (Fig. 4, a, b, and j; and Fig. S3, d and e). Since MRCK can activate myosin-II, the defect in pseudopod remodeling during normal cup closure agrees with a function of myosin-II–dependent contractility in limiting actin polymerization (Reymann et al., 2012). The prolonged pseudopods failed to encircle POS, which remained misaligned, supporting a role for MRCKβ-activated contractility in cup maturation and POS engulfment (Fig. 4, a, b, and j; and Fig. S3, d and e). To further confirm that MRCKβ and myosin-II function in the same internalization pathway, we compared the effects of single or combined treatments with blebbistatin or MRCKβ siRNA. Single or combined inhibition of MRCKβ and myosin-II resulted in similar levels of F-actin–rich pseudopod overgrowth and POS internalization defects (Fig. 4, q–v). Together with the observation that recovery from blebbistatin washout requires MRCK activity (Fig. 3, n–p), these results indicate that RPE phagocytosis requires MRCKβ-mediated myosin-II activation for cup morphogenesis.

## N-WASP is required for protrusion induction and particle wrapping in the RPE

In vitro, F-actin polymerization and the formation of actin network architecture define myosin motor activity (Reymann et al., 2012). Since N-WASP contributes to protrusion induction, we next determined the relationship between Cdc42-driven N-WASP and MRCKβ signaling. Inhibition of N-WASP using Wiskostatin or its siRNA-mediated knockdown drastically reduced the number of cells forming normal cups (Fig. 5, a–i), whilst POS binding to the apical membrane of RPE cells was unaffected. Reduced levels of protrusion induction were still observed, possibly reflecting actin polymerization stimulated by

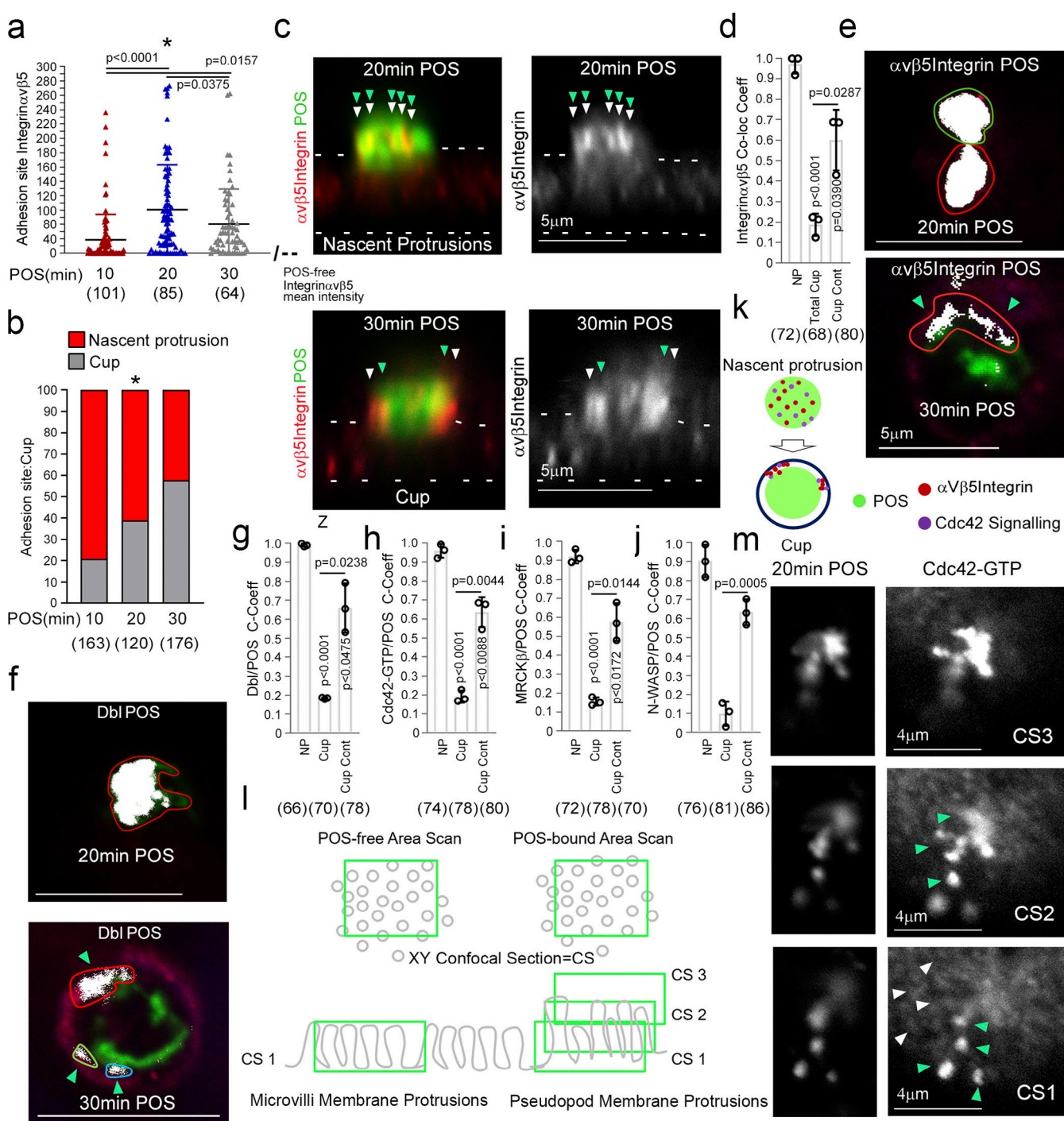

Figure 2. **POS binding induces rapid recruitment of Cdc42 signaling components. (a–j)** Integrin αvβ5 and Cdc42 signaling components recruitment to POS adhesion sites upon 10 (early adhesion), 20 (membrane deformation and protrusions), and 30 min (cups). Note, POS binding to the integrin αvβ5 receptor induces rapid colocalization of Dbl3, Cdc42-GTP, MRCKβ, and N-WASP with the POS–integrin complex. Nascent protrusions (white arrowheads) display almost complete colocalization of Cdc42-signaling components with POS (highlighted by green arrowheads) whereas in the cup conformation (white arrowheads) colocalization occurs at discrete contact sites (green arrowheads) at the cup–POS interface. **(k)** Schematic diagram illustrating almost complete co-localization of integrinαvβ5 and Ccd42-signaling proteins with POS at nascent protrusions and partial co-localization at POS-cup adhesion sites. **(l)** Schematic diagram illustrating method of quantifying mean labeling intensity of protrusion proteins using Cdc42-GTP as an example. Confocal *xy* sections were scanned at the apical membrane surface of RPE cells (Confocal Scan 1, CS1), both unattached and POS attached. Since protrusions are at an increased height compared to normal apical brush border microvilli, further serial scans were taken as appropriate to cover the protrusion length and values were averaged. **(m)** At CS1, Cdc42-GTP enrichment is observed at single protrusions (green arrowheads) at a similar diameter to surrounding non-POS attached brush border structures (white arrowheads). Quantifications: means ±1SD of *n* = 3 independent experiments; indicated are the total number of cells analyzed and P values derived from *t* tests, except for the scatter plot in panel a where the median and upper and lower quartiles are highlighted, and *n* values represent number of cells collected from three independent experiments, and significance was tested with an ANOVA test.

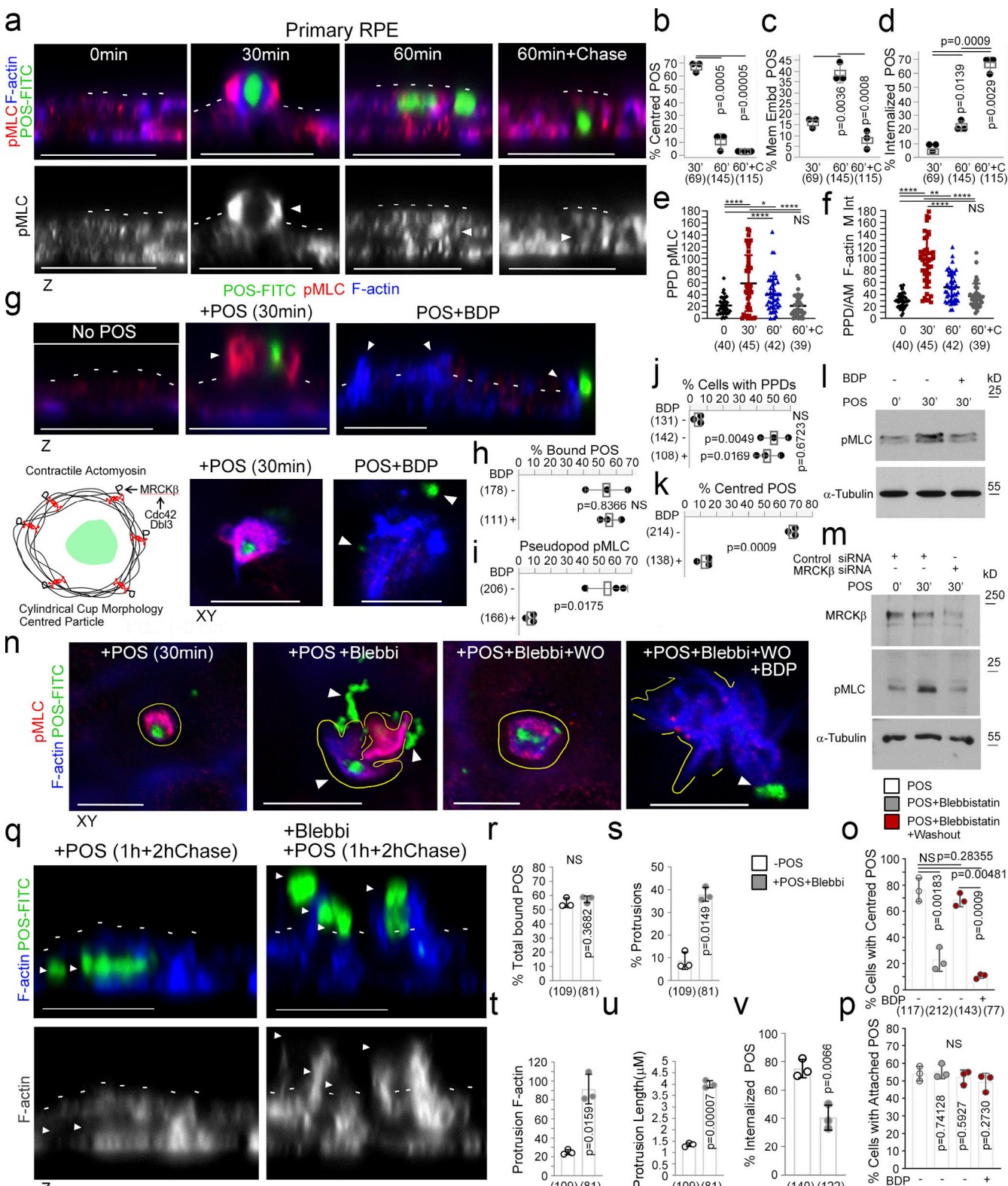

Figure 3. **MRCKβ regulates actomyosin dynamics to control POS wrapping in the RPE. (a–f)** Time course of POS-induced cup maturation, closure, and POS internalization reveals myosin motor activation and corresponding F-actin enrichment peaks with membrane cups wrapping around POS. **(g–m)** POS adhesion to porcine primary RPE cells after 30 min of incubation results in mature cups wrapping around POS that are enriched with pMLC, whilst MRCKβ inhibition prevents myosin activation and cup remodeling around POS. **(n–p)** Inhibition of myosin-II motor activity using blebbistatin results in similar membrane remodeling and POS centering defects to MRCKβ inhibition that is reversed within 2 h following blebbistatin washout when MRCKβ is active. **(q–v)** Binding of POS to blebbistatin treated cells results in pseudopod overgrowth and inhibition of POS internalization. Protein staining was quantified as mean intensity. Scale bars represent 10 μm, unless highlighted otherwise. All white dashes highlight the apical F-actin cortex. White arrowheads highlight

F-actin-rich protrusions and/or POS position. PPD, pseudopod; BDP, BDP5290 MRCK inhibitor; Mem Embd, membrane embedded; AM, apical membrane; M Int, mean intensity. All quantifications are based on $n$ = 3 independent experiments. Shown are the data points, means ±1SD, the total number of cells analyzed for each type of sample across all experiments, and P values derived from t-tests, except for e and f scatter plots where the median and upper and lower quartiles are highlighted, and $n$ values represent number of cells collected from three independent experiments and significance based on an ANOVA test (****, P < 0.0001; **, P < 0.006; *, P < 0.03). Source data are available for this figure: SourceData F3.

the previously described integrin αvβ5–Rac pathway (Mao and Finnemann, 2015). These protrusions contained less F-actin (~50%; Fig. 5, c–e and g–i). Hence, N-WASP is required for full protrusion and cup induction.

MRCK inhibition resulted in pseudopod overgrowth, suggesting defective regulation of actin polymerization (Figs. 3 and 4). To confirm the involvement of actin polymerization, Latrunculin A (Coue et al., 1987) was added to MRCKβ-depleted cells, which resulted in inhibition of protrusion overgrowth (Fig. 5, j–n). siRNA-mediated knockdown of both N-WASP and MRCKβ also inhibited F-actin–rich protrusions overgrowth when compared with MRCKβ depletion alone (Fig. 5, j–n). These results indicate that N-WASP–stimulated F-actin polymerization regulated by MRCKβ activity is required for normal phagocytic cup formation (Fig. 5 o).

## MRCKβ drives FAK activity and recruitment of force transducers

Protrusions transform into mature cups with increased F-actin accumulation and myosin-II activity (Fig. 6, a–f and Fig. S3 j), indicating increased mechanical force may act on the phagocytic cup to drive cup closure and internalization. Strikingly, there was a progressive increase in the recruitment of talin and vinculin as protrusions matured into cups, correlating with F-actin and pMLC accumulation (Fig. 6, g–j). MRCKβ colocalized with the mechanosensing machinery at cups in clusters (Fig. 6, k and l). Wiskostatin and BDP5290 reduced talin cluster labeling intensity at cups (Fig. 6 m and Fig. S3 f). At focal adhesions, vinculin and talin form a molecular clutch that transduces mechanical force to integrins, a function regulated by FAK (Chen, 2008; Oria et al., 2017). Inhibition of MRCKβ or myosin-II inhibited FAK-Y397 phosphorylation in the presence of POS, which occurs after 30 min in control cells and correlates with cup formation (Fig. 6 n). siRNA-mediated knockdown of MRCKβ also inhibited FAK-Y397 phosphorylation in the presence of POS (Fig. 6 o). Thus, MRCKβ-stimulated myosin-II activation stimulates FAK activation and recruitment of mechanical force transducers to POS adhesion sites.

Cytoskeletal tension acting on receptors induces clustering to strengthen particle adhesion (Oria et al., 2017; Sobota et al., 2005). Inhibition of MRCKβ activity did not affect initial accumulation of αvβ5 integrins at nascent protrusions in agreement with the low level of actomyosin contractility observed at this early stage (Fig. S3 g). However, integrin clustering at mature cups was inhibited by MRCK inhibition (Fig. 6, p and q, and Fig. S3 h) or siRNA-mediated knockdown (Fig. 6, r and s, and Fig. S3 i). Therefore, increasing MRCKβ-stimulated actomyosin activity drives phagocytic cup maturation and leads to the activation of mechanosensitive signaling and integrin clustering prior to internalization (Fig. 6 t).

## MRCKβ is required for RPE function in vivo

We next asked whether MRCKβ is required for retinal function in vivo. To test this, we deleted MRCKβ in the RPE of adult mice, a nonproliferative polarized epithelium, by subretinal injection of lentiviral CRISPR-Cas9 vectors, a method that enables specific transduction of the RPE (Fig. 7, a and b, and Fig. S4 a; Georgiadis et al., 2010). Either of two lentiviral CRISPR-Cas9 vectors targeting different regions of the MRCKβ gene resulted in RPE-specific knockout of MRCKβ (Fig. S4, d and e). Analysis of retinal tissue sections by confocal microscopy revealed that mice maintained for 21 d after injection displayed a striking thinning of the outer nuclear layer (ONL), a hallmark of advanced retinal degeneration (Fig. S4 b; Edwards and Szamier, 1977). At 7 d after injection, the ONL thickness and outer limiting membrane F-actin intensity were unchanged (Fig. 7, c and d), indicating that the general architecture and integrity of the retina were maintained. However, MRCKβ knockout resulted in reduced apical–basal F-actin intensity ratio (Fig. 7, c and d; and Fig. S4, f–h). The apical RPE membrane in vivo is covered by extended microvilli that function in phagocytosis (Bonilha et al., 2006); hence, we investigated the ultrastructure of the RPE using TEM. In control RPE cells, apical microvilli were ordered and packed into linear arrays that extended to and surrounded POS, as previously reported (Fig. 7, h and j; Bonilha et al., 2006). RPE cells contained internalized phagosomes, indicating functional RPE (Fig. 7, h and j). In contrast, microvilli of MRCKβ-knockout RPE in contact with POS appeared disordered and more spread out (Fig. 7, i and k). Measurement of microvilli thickness and packing into sheets revealed a decrease in microvilli packing order (Fig. 7, l and m). Strikingly, MRCKβ-targeted RPE cells contained >7 times less cytoplasmic phagosomes, which was paralleled by an accumulation of extracellular shed POS (Fig. 7, e–g). Thus, MRCKβ is required for RPE phagocytosis and retinal integrity in vivo.

## MerTK activates Dbl3 to drive phagocytosis in the RPE

We next asked whether the upstream GEF of MRCKβ, Dbl3, is activated by MerTK to stimulate phagocytosis. As αvβ5 integrin and the Dbl3-activated Cdc42 signaling machinery, MerTK was enriched at nascent protrusions and almost completely colocalized with POS (Fig. 8, a–c, and Fig. S5, a and b). In mature cups, integrin αvβ5 and Cdc42 signaling components only significantly colocalized with POS at POS-membrane contacts; in contrast, MerTK still displayed significant colocalization (Fig. 8 c and Fig. S5, a–e). Higher colocalization of MerTK with POS was expected as POS binding promotes cleavage of the extracellular N-terminal domain that then quenches still available MerTK binding sites on POS (Law et al., 2015). Activated tyrosine kinase pSrcY416, known to be activated by MerTK (Shelby et al., 2015), also localized to POS-induced protrusions (Fig. 8 b and Fig. S5 f).

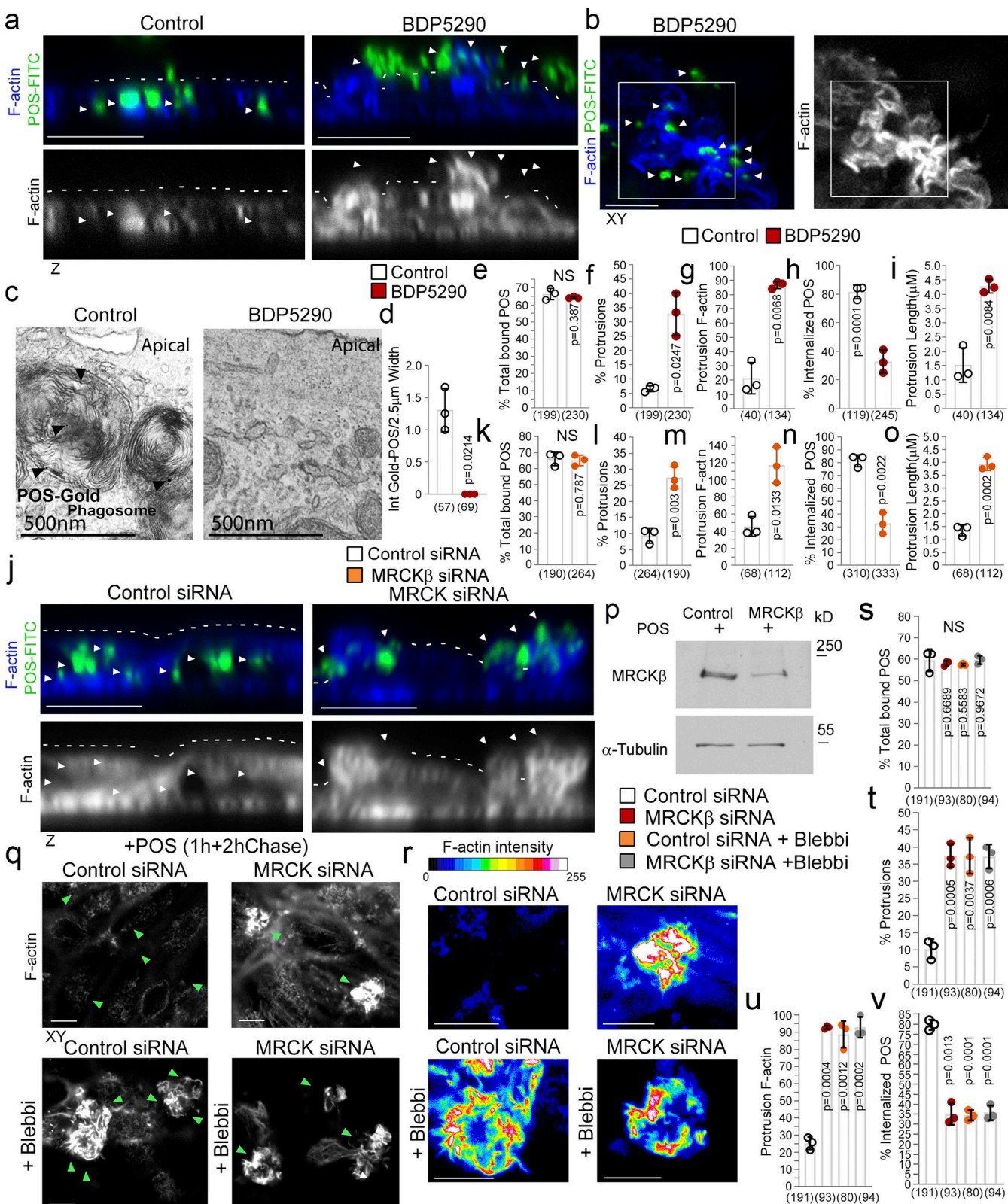

**Figure 4. MRCKβ and myosin-II control cup morphogenesis and POS internalization. (a–i)** Inhibition of MRCKβ activity in primary porcine RPE cultures results in defective POS-induced apical membrane remodeling with stalled F-actin–based protrusions and inhibited phagocytosis as determined by confocal z-sections of porcine primary RPE cells stained with Atto647-Phalloidin; POS were labeled with FITC. White dashes highlight apical membrane and arrowheads internalized POS (left panels) or extracellular POS (right panels). TEM of primary porcine RPE cells using Gold-labeled POS confirmed MRCK kinase inhibition using BDP5290 results in an inhibition of phagocytosis. **(j–p)** Knockdown of MRCKβ resulted in similar defects in cup morphogenesis and phagocytosis. **(q–v)** Inhibition of myosin-II motor activity in MRCKβ-depleted cells results in identical defects in cup morphogenesis and phagocytosis, confirming that both

defects are due to a single pathway. Note, heat maps signify F-actin intensity, white and green arrowheads highlight the position of POS, and all scale bars represent 10 μm. The quantifications in d–h are based on $n = 3$ independent experiments and show data points, means ±1SD (top of bar), and the total number of cells. P values were derived from $t$ tests. Source data are available for this figure: SourceData F4.

To test whether Dbl3-activated actomyosin dynamics in response to POS require MerTK activation, primary porcine RPE cells were cultured in serum-free medium to eliminate effects of serum MerTK ligands. The cells were then stimulated with POS-FITC in the absence or presence of Gas-6, the ligand that links MerTK to POS (Etienne-Manneville and Hall, 2002; Hall et al., 2002; Kevany and Palczewski, 2010), without or with N-WASP or MRCK inhibitors. POS binding to the apical membrane was observed without addition of Gas-6, reflecting binding to the integrin receptor, but the induction of protrusions was inhibited (Fig. 8, d and e). Addition of Gas-6 together with POS induced an increase in protrusions positive for pMLC (Fig. 8, d and e). Inhibition of N-WASP prevented protrusion formation, and inhibition of MRCKβ prevented pMLC induction and cup-POS centering (Fig. 8, e–g). Knockdown of Dbl3 attenuated pMLC-positive and POS wrapping, supporting a role for the GEF downstream of MerTK and upstream of MRCKβ (Fig. 8, h and i).

Localization of the MerTK effector c-Src at nascent protrusions along with MerTK and Dbl3 suggested that Dbl3 activation may involve tyrosine phosphorylation. Immunoprecipitation of phosphotyrosine-containing proteins in primary porcine RPE cells stimulated with POS for 20 min revealed tyrosine phosphorylation of Dbl3 (Fig. 8 j). Bioinformatics analysis using NetPhos3.1 suggested a conserved tyrosine phosphorylation motif TERVYVREL in the Dbl-homology (DH) domain that carries GEF activity (Fig. 8 k and Fig. S5 g). This motif had been shown to be phosphorylated in Dbl1, a functionally distinct splice variant of Dbl3 that lacks apical targeting information but contains a homologous DH domain (Gupta et al., 2014; Zihni et al., 2014). The TERVYVREL motif of Dbl1 is a Src family kinase target, and phosphorylation activates its GEF activity in vitro (Gupta et al., 2014). Src tyrosine kinase inhibition blocked Dbl3-myc driven phagocytosis in ARPE19 cells (Fig. 8, l and m). Conversely, expression of a phosphomimetic Dbl3Y570D mutant increased internalization. Thus, Dbl3 activation of MRCKβ and subsequent control of phagocytosis requires Src activity (Fig. 8 n).

### MRCKβ signaling regulates phagocytosis in macrophages
We next tested the involvement of MRCKβ in FcR-mediated macrophage phagocytosis, which is Cdc42-dependent (Caron and Hall, 1998; Massol et al., 1998). In macrophages differentiated from THP-1 cells, MRCKβ associated with the cell cortex (Fig. S5 h). Fluorescent, IgG-opsonized Zymosan-555–labeled yeast particles were efficiently internalized into the cell body by control siRNA-treated cells, as determined by confocal z-section analysis, but not cells treated with siRNAs targeting MRCKβ, where ~60% of cell-associated particles appeared tethered to the cell cortex (Fig. 9, a–d, and Fig. S5 i). The role of myosin-II, the substrate of MRCKβ (Unbekandt et al., 2014; Zhao and Manser, 2015; Zihni et al., 2017), in FcR-mediated phagocytosis is conflicting with recent studies indicating that it may not

be essential (Barger et al., 2020; Rotty et al., 2017). We found that the efficiency of particle internalization as determined by measuring both total zymosan-555 and anti-IgG-488 labeled external particles in nonpermeabilized cells, was not significantly affected following MRCKβ knockdown, agreeing with recent studies that internalization is not myosin-dependent (Fig. 9, e and f). However, our results indicate that MRCKβ signaling promotes efficient transfer of internalized zymosan particles from the macrophage cell cortex to the cell body. The distinct employment of cortical Cdc42/MRCKβ signaling by different phagocytic machineries in the RPE and macrophages indicates plasticity of the signaling module, enabling the Cdc42/MRCKβ mechanism to be adapted to the requirements of different phagocytic receptors and mechanisms.

### Dbl3Y570D rescues phagocytosis in MerTK-deficient RPE
Dbl3 activates MRCKβ and its coeffector N-WASP to drive phagocytosis following POS-MerTK ligation; hence, enhancing Dbl3 signaling may be sufficient to stimulate POS internalization in phagocytosis-deficient RPE. To test this, we employed RPE cells generated from iPSCs derived from a retinitis pigmentosa individual carrying biallelic nonsense mutations in the MerTK gene, leading to loss of MerTK protein expression, phagocytosis deficiency, and blindness (Fig. 10, a–d, and Fig. S5 j; Ramsden et al., 2017). MerTK-deficient iPSC-derived RPE cells differentiate and polarize normally with functional integrin αvβ5 receptors, except for the absence of MerTK protein (Ramsden et al., 2017). As previously reported, these cells bound POS efficiently but displayed poor phagocytosis after exposure to POS-FTIC for 3 h (~11%; Fig. 10, e–g). Endogenous Dbl3 colocalized with attached POS characteristic of early adhesion sites with no significant protrusion or cup formation detected, indicating a lack of Dbl3-induced signaling in agreement with a model in which MerTK signaling activates Dbl3 (Fig. 10, h and i; and Fig. S5, i, k, n, and o). Expression of Dbl3-myc in MerTK-deficient RPE increased phagocytosis twofold (~22%; Fig. 10 g and Fig. S5 l). Expression of the phosphomimetic Dbl3Y570D mutant increased phagocytosis to 87.0% and promoted efficient maturation of POS adhesion sites with characteristic Dbl3 colocalization at the POS periphery (Fig. 10, e, g, h, j, and k; and Fig. S5, m and p–s). Thus, Dbl3Y570D expression functionally mimics MerTK signaling and rescues phagocytosis in MerTK-deficient RPE cells. Stimulation of the MRCKβ pathway by Dbl3 is therefore sufficient to rescue phagocytosis in MerTK-deficient cells, indicating that it represents the main phagocytic effector pathway of MerTK.

## Discussion
This study identifies an MRCKβ-driven signaling mechanism that spatiotemporally regulates actomyosin contractility to guide distinct morphological membrane transformations

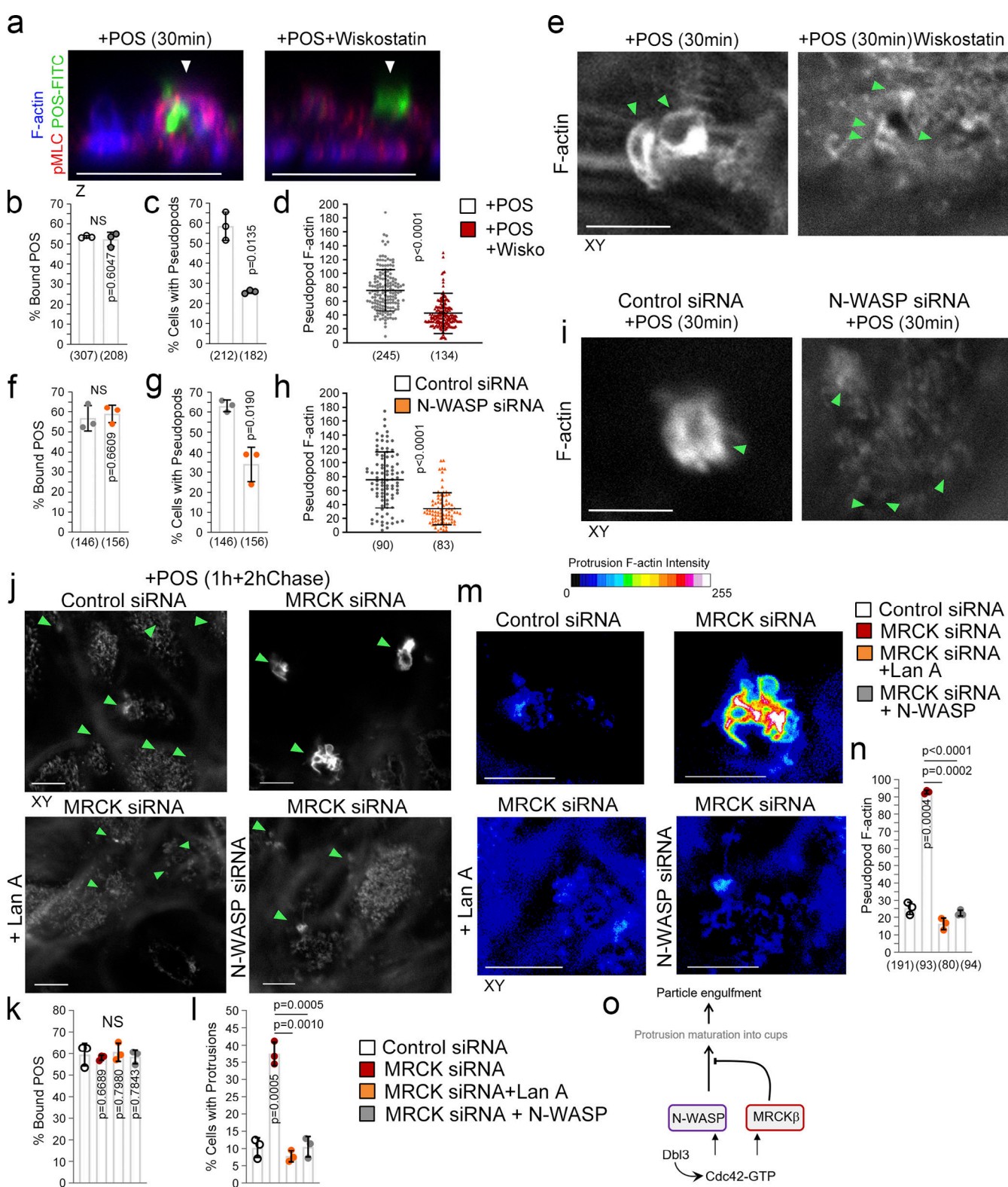

Figure 5. **N-WASP is required for cup formation. (a–i)** Inhibition of N-WASP activity attenuates protrusion induction and cup maturation. Green and white arrowheads highlight POS adhesion sites. **(j–n)** Inhibition of actin polymerization using the inhibitor Latrunculin A or siRNA of N-WASP prevents F-actin–rich protrusion overgrowth due to MRCKβ depletion. Green and white arrowheads highlight POS adhesion sites. **(m and n)** Heat maps signify F-actin intensity. **(o)** Schematic figure illustrates a model in which Dbl3 activates Cdc42 at POS-membrane adhesions to drive N-WASP–dependent cup formation that is regulated by MRCKβ to control the size and shape and wrapping of POS as nascent protrusions mature into cups. Protein staining was quantified as mean intensity. Scale bars represent 10 µm, unless highlighted otherwise. All quantifications are based on n = 3 independent experiments. Shown are the data points and means ±1SD for b, c, f, g, k, l, and n, scatter plots for d and h, where the median and upper and lower quartiles are highlighted represents the total number of cells analyzed for each type of sample across all experiments, and P values derived from t tests.

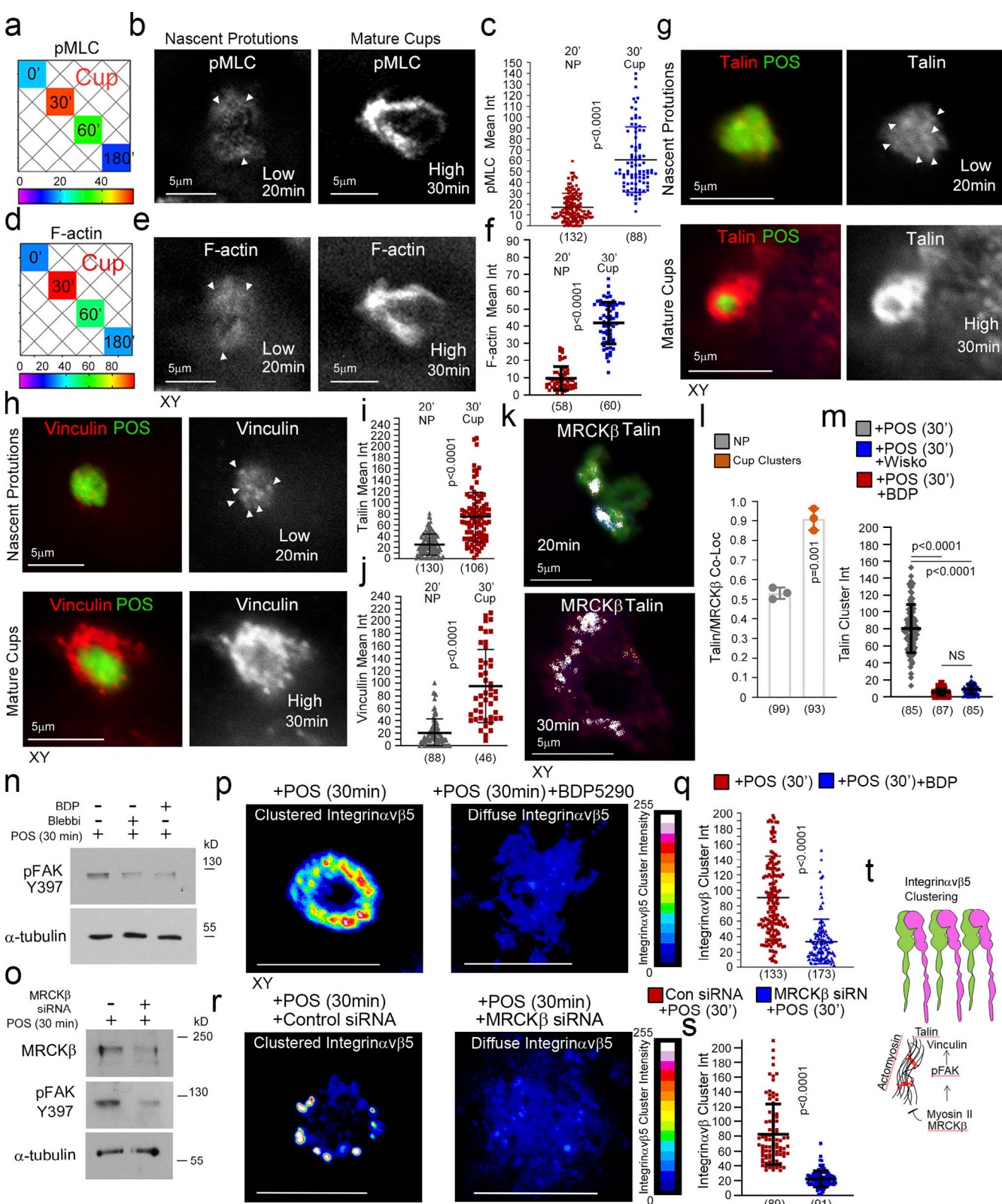

Figure 6. **MRCKβ regulates mechanotransduction. (a)** Heat map of POS-RPE incubation time course (data extracted from Fig. 3, e and f). Note, both pMLC and F-actin labeling peak at cup formation. **(b–f)** Both F-actin and pMLC staining are weak during the induction of nascent protrusions (highlighted by white arrowheads) and increase sharply during cup formation and morphogenesis. **(g–j)** Confocal microscopy analysis of primary RPE treated with POS-FITC revealed recruitment of mechanosensors talin and vinculin at maturing cups concomitant with robust actin polymerization and increased actomyosin contractility. White arrowheads highlight weak localization of proteins at nascent protrusions. Values were calculated by subtracting mean staining intensity of the apical membrane that is not attached to POS in the same population. **(k and l)** MRCKβ colocalizes with mechanosensory talin in maturing cups, in clusters.

**(m)** Use of wiskostatin and BDP inhibitors block clustering of mechanosensory talin. **(n and o)** Inhibition of actomyosin contractility, during cup maturation, by inhibiting myosin-II motor activity using blebbistatin, or MRCKβ kinase activity using BDP, or siRNA knockdown of MRCKβ inhibits FAK phosphorylation. **(p–s)** Inhibition of MRCK-driven actomyosin contractility using BDP or siRNA knockdown of MRCKβ inhibits clustering of αvβ5 integrin at cups. Note, αvβ5 integrin appears diffuse at cups along with loss of cup morphology and POS centering. Protein staining was quantified as mean intensity (int). **(t)** Schematic diagram illustrating the dual function of MRCKβ driven contractility that limits actin polymerization to control maturation of protrusions into cups, whist activating FAK to drive molecular clutch assembly. Scale bars represent 10 μm, unless highlighted otherwise. Shown are scatter plots where the median and upper and lower quartiles are highlighted, *n* represents the total number of cells analyzed for each type of sample across all experiments, and P values derived from *t* tests except for m where P values are derived from ANOVA tests. In the bar graph (l). shown are the data points and means ±1SD. P values were derived from *t* tests. Source data are available for this figure: SourceData F6.

required for RPE phagocytosis. This mechanism functions downstream of MerTK, a common receptor generally found on phagocytes that internalize dead cells and cell remnants such as POS in the retina. Expression of a mutant form of the Cdc42 exchange factor Dbl3 that mimics MerTK-activation is sufficient to rescue phagocytosis in MerTK-deficient RPE cells, indicating that it represents the principal phagocytic effector pathway activated by MerTK.

In the RPE, MRCKβ is activated by MerTK-stimulated phosphorylation of the apical Cdc42 GEF Dbl3 to drive myosin-II–mediated actomyosin contractility to guide maturation of particle-induced protrusions into cups in cooperation with N-WASP. POS-membrane adhesion results in a rapid enrichment of the receptors αvβ5 integrin and MerTK at adhesion sites along with the Cdc42 signaling machinery, resulting in localized MerTK-activated Cdc42 signaling. This stimulates actin polymerization, at least in part due to N-WASP, which forms the basis for generating membrane protrusions. MRCKβ-stimulated contractility then controls maturation of protrusions into cups and particle wrapping by regulating actomyosin dynamics. Inactivation of MRCKβ results in extended growth of pseudopods, indicating that MRCKβ-regulated contractility is required to limit actin polymerization. These findings are in agreement with a model in which the interdependence of actin polymerization and contractility creates the conditions for the spatial and temporal control of actin cytoskeletal dynamics required to drive complex morphodynamic processes (Reymann et al., 2012).

MRCKβ-mediated myosin-II activity proceeds to guide particle engulfment and contractile force-dependent pulling into the cell by coupling to additional cytoskeletal mechanisms. During the transition from particle wrapping to engulfment, MRCKβ-activated contractility promotes formation of a force-transducing bridge between integrins and accumulating actin fibers by activation of FAK, stimulating recruitment of vinculin and talin, as well as integrin clustering. αvβ5 Integrin has been shown to activate FAK at an earlier step, on initial binding to POS, which has been proposed to contribute to the rhythmic burst activity of RPE phagocytosis in vivo (Finnemann, 2003; Finnemann et al., 1997). Rac1 is activated by αvβ5 integrin, although its precise role in phagocytosis is unclear since its activity is independent of FAK and MerTK (Mao and Finnemann, 2012). Rac1 may contribute to the remaining protrusion formation observed in N-WASP depleted cells, indicating cooperative mechanisms driving actin polymerization. Cdc42 and Rac1 have been shown to display spatiotemporally coordinated localization and morphogenic transformation profiles in FcR-mediated macrophage engulfment (Hoppe and Swanson, 2004); thus, future

work to determine the mechanistic relationship between the two GTPases is likely to provide important information about coordination of their functions in phagocytosis. As FAK activation by αvβ5 integrin first enhances MerTK activity and FAK activity stimulated by MRCKβ then functions later most likely to promote recruitment of mechanotransducers and integrin clustering, our results suggest a biphasic activation of FAK. FAK activation by MRCKβ is reminiscent of mesenchymal migration, where myosin-II–dependent FAK-mediated phosphorylation of paxillin promotes vinculin binding to talin and actin to reinforce the force transducing link between the cytoskeleton and the ECM (Chen, 2008; del Rio et al., 2009; Humphries et al., 2007; Pasapera et al., 2010; Thievessen et al., 2013). Since spreading of migratory cells on a matrix is topologically similar to phagocytosis of particles (Grinnell, 1984), our results support a model that postulates the existence of signaling mechanisms that control the actomyosin cytoskeleton during phagocytosis in a similar manner as that during cell spreading.

Complement receptor-mediated macrophage phagocytosis, which is myosin-II–independent, also stimulates recruitment of such "molecular clutch" proteins, suggesting that force generation during cup morphogenesis in macrophages can be based on distinct mechanisms (Jaumouille et al., 2019). The role of myosin-II in FcR-mediated internalization is controversial and may not involve a direct role in cup formation (Kovari et al., 2016; Rotty et al., 2017). Since myosin II activity increases cortical tension, a property that propels particles inward (Gandhi et al., 2009), it has been suggested that myosin-II may facilitate particle inward movement (Jaumouille and Waterman, 2020). However, the dissection of the role of myosin-II has been challenging, and to complicate matters further, myosin function in some processes may be motor-independent (Choi et al., 2008), although such processes are probably still due to a generation of contractility and tension (Ma et al., 2012). Our results obtained upon inhibiting MRCK indicate that it plays a role important for particle translocation from the cortex to the cell body after internalization. This contrasts with the role of MRCK in this signaling mechanism in the RPE. Thus, while Cdc42/MRCKβ signaling is required downstream of different phagocytic receptors, it regulates the process from early steps starting with particle engulfment during MerTK-regulated phagocytosis in the RPE, but only late steps postinternalization in FcR-mediated phagocytosis in macrophages, indicating that plasticity of the signaling machinery enables adaptation to the specific requirements of different phagocytic receptors and cell types.

The essential function of MRCKβ-controlled morphogenetic signaling in distinct receptor-mediated phagocytes and

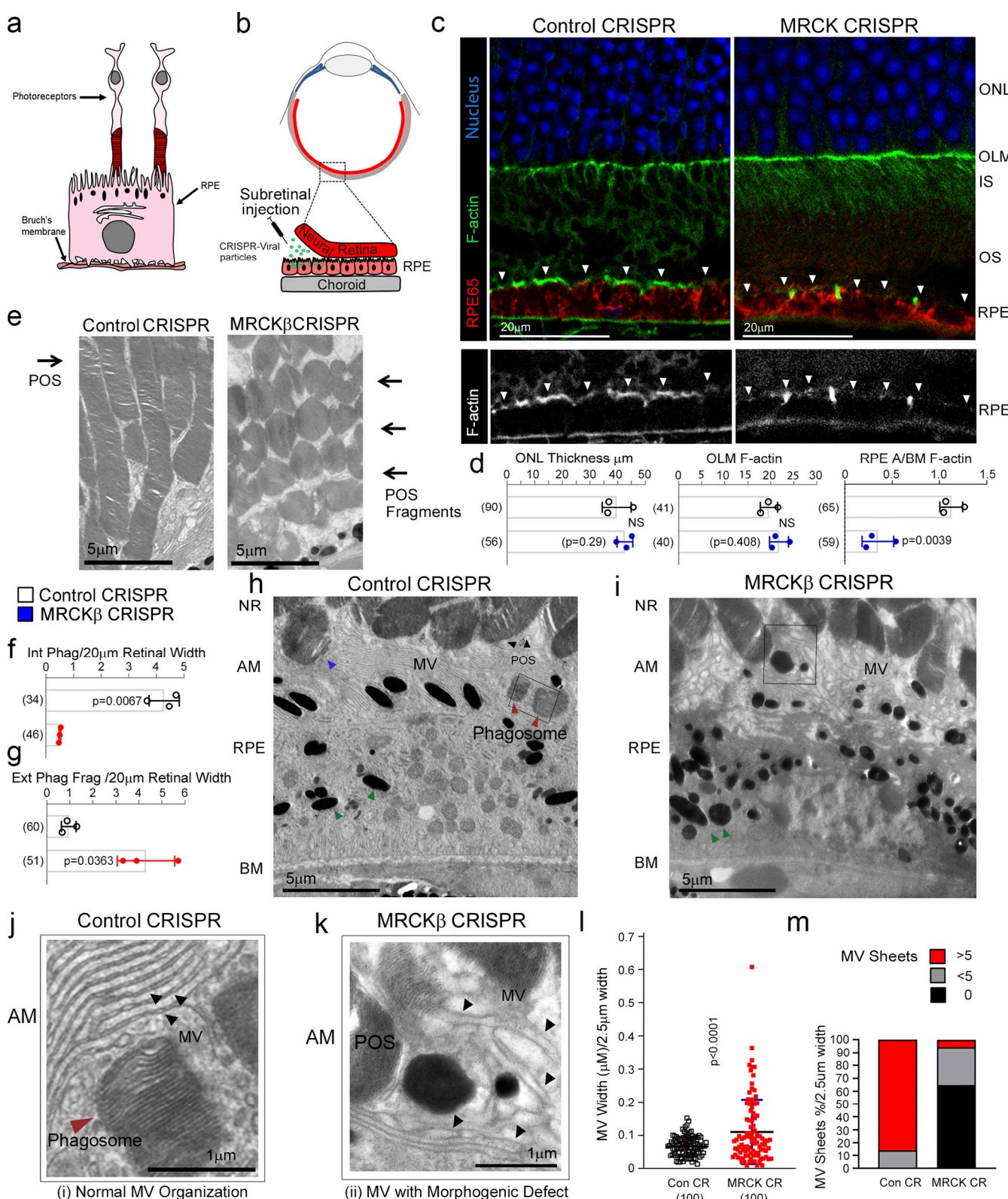

Figure 7. **MRCKβ is required for RPE function and retinal integrity in vivo. (a)** Schematic illustration of RPE and photoreceptor architecture from the retina. Note, POS are in contact with microvilli at the apical membrane of RPE. As POS is shed at the tips, the resultant fragments expose membrane components that initiate the phagocytic cycle. **(b)** Schematic illustration of subretinal delivery of lentiviral vectors. Note, injection of viral particles into the subretinal space results in specific transduction of the RPE layer. **(c and d)** Confocal immunofluorescence analysis of retinal sections from mice injected with control or MRCKβ knockout vectors reveal no significant difference in the overall retinal structure after 7 d but reduced apical F-actin staining, with reference to basal F-actin, in the MRCKβ–deficient RPE. White arrowheads highlight the apical F-actin cortex. **(e–i)** TEM analysis shows that RPE from control CRISPR-vector-injected mice internalized POS fragments whereas internalization in MRCKβ CRISPR-vector injected mice was inhibited, resulting in accumulation in the

extracellular space. Red and blue arrowheads highlight internalized and internalizing POS fragments, respectively, in control RPE. Green arrowheads highlight melanosomes in both control and MRCKβ knockout samples. **(j–m)** TEM analysis of RPE apical microvilli architecture in control and MRCKβ CRISPR-vector-injected mice. Dashed boxes in h and i signify the location of areas zoomed in. Note, the quantifications show that microvilli from controls display ordered sheet packing and individual morphology, whereas from MRCKβ knockout animals display a disordered appearance of microvilli with a loss of the sheet arrangement and irregular individual thickness. A/BM, apical to basal membrane ratio; AM, apical membrane; NR, neural retina; MV, microvilli; OLM, outer limiting membrane. Black arrowheads highlight individual mv strands. Quantifications in bar graphs show means ± 1SD, $n = 3$. In l, a column scatter plot from three independent experiments highlights the median and upper and lower quartiles. P values are derived from $t$ tests.

its importance for retinal integrity indicate that its associated mechanism may have a use as a therapeutic target in disease. As Dbl3 is activated by POS binding to MerTK, receptor-independent activation of Dbl3 signaling should stimulate phagocytosis. Indeed, expression of phosphomimetic Dbl3 in retinitis pigmentosa iPSC-derived MerTK-deficient RPE cells rescued phagocytosis, indicating that it represents the main phagocytic effector pathway downstream of MerTK. The ability of Dbl3 to rescue phagocytosis indicates that its localization depends on other factors that may include ezrin and its Sec14-like domain, both factors important for apical and membrane localization of Dbl3 (Zihni et al., 2014). The ability to rescue phagocytosis despite loss of phagocytic receptor function may lead to broader therapeutic applications in the RPE beyond MerTK dysfunction, including an age-related decline in phagocytosis, which may contribute to age-related macular degeneration, a leading cause of blindness (Inana et al., 2018; Sparrow et al., 2010). MerTK-regulated phagocytosis is important for the function of diverse phagocytes, and its deregulation is thought to contribute to difficult-to-treat diseases such as neurodegeneration (Brosius Lutz et al., 2017; Haukedal and Freude, 2019; Myers et al., 2019). Similarly, macrophages and other phagocytic cells of the immune system play important roles in infection, chronic inflammatory disease, and cancer, and can promote tissue repair as well as damage. Therefore, understanding the molecular mechanisms that control phagocytosis and identifying new approaches to modify such mechanisms will contribute to designing innovative new therapies for a wide range of diseases (Ardura et al., 2019).

## Materials and methods
### Cell culture and generation of mammalian expression and viral vectors
Human ARPE-19 cells were obtained from ATCC. Primary porcine RPE cells were isolated from pig eyes as described previously (Tsapara et al., 2010). ARPE-19 and porcine RPE cells were grown in DMEM supplemented with 10% FBS. Once the isolated porcine RPE cells were confluent, they were cultured in DMEM medium containing 1% FBS and passaged no more than two times. For phagocytosis assays, the cells were transferred to medium containing 2.5% FBS. Fresh batches of ARPE-19 cells from a contamination-free stock that had been tested for mycoplasma (MycoAlert, Promega, Inc.) were used every 6–8 wk. Cells were then weekly stained with Hoechst dye to reveal nuclei and DNA of contaminants such as mycoplasma. iPSC cells carrying inactivating mutations in MerTK were characterized previously and were differentiated into RPE cells as described (Ramsden et al., 2017). MRCK inhibitor BDP5290 was generously

provided by Michael Olson (Cancer Research UK Beatson Institute, Glasgow, UK) and was synthesized by the Cancer Research UK Beatson Institute Drug Discovery Group. It was used at a concentration of 10 mM. Blebbistatin (used at 10 µM or 40µM) and Wiskostatin (used at 5 µM) were purchased from Tocris Bioscience. The expression plasmid for Dbl3-myc was as described (Zihni et al., 2014). Dbl3Y570D substitution was introduced into Dbl3-myc by PCR using the CloneAmp HiFi PCR premix and In-Fusion cloning (Clontech, Inc.). Constructs were verified by sequencing. CRISPR-Cas9 lentiviral vectors were obtained from Sigma-Aldrich and were constructed in pLV-U6g-EPCG. The sequences targeted were for MRCKβ 5′-CCTGGACGG GCCGTGGCGCAAC-3′ and 5′-ACACCGAGTGCAGCCACTCGG-3′, and for Dbl3 5′-TGGAGATCGAAGACTGGATACATGG-3′. Plasmids were transfected using TransIT-X2 (Cambridge Bioscience).

### Differentiation of patient-derived iPSCs into RPE
Patient-derived iPSCs were cultured and differentiated into RPE as previously described (Ramsden et al., 2017). Briefly, isolated iPSC colonies were cultured on hESC-qualified Matrigel (354277; BD Biosciences) in Essential 8 or Essential Flexi with supplements (Thermo Fisher Scientific). Once ~80% confluent, the media was changed to differentiation media (Vugler et al., 2008) and fed twice weekly for at least a further 8 wk until pigmented foci of differentiated RPE were observed. These pigmented foci were isolated manually and purified and fed every 3–4 d with X-VIVO 10 medium (LZBE 04-743Q; Lonza) with gentamicin (Thermo Fisher Scientific).

### Transfection
Cells were cultured and transfected with Lipofectamine RNAiMAX (Thermo Fischer Scientific) or Vitromer Blue (Lipocalyx) using siRNAs targeting the following sequences: human N-WASP 5′-CAGAUACGACAGGGUAUCCAA-3′ and 5′-UAGAGAGGGUGCUCAGCUAAA-3′; porcine Cdc42, 5′-GAUGAC CCCUCUACUAUUG-3′; porcine Dbl3 5′-AAGACAUCGCCUUCC UGUC-3′ and 5′-AUACCUGGUCUUCUCUCAA-3′; and porcine MRCKβ 5′-CGAGAAGACUUCGAAAUAA-3′ and 5′-AGAGAA GACUUUGAAAUAU-3′. The nontargeting control siRNAs and those for human Cdc42, MRCKβ, and Dbl3 were as described previously (Zihni et al., 2014; Zihni et al., 2017).

### Maintenance of mice
Wild-type mice (C57BL/6J) were purchased from Harlan Laboratories (Blackthorn). All mice were maintained under cyclic light (12 h light–dark) conditions; cage illumination was 7-foot candles during the light cycle. All experiments were approved by the local Institutional Animal Care and Use Committees (University College London [UCL]) and conformed to the

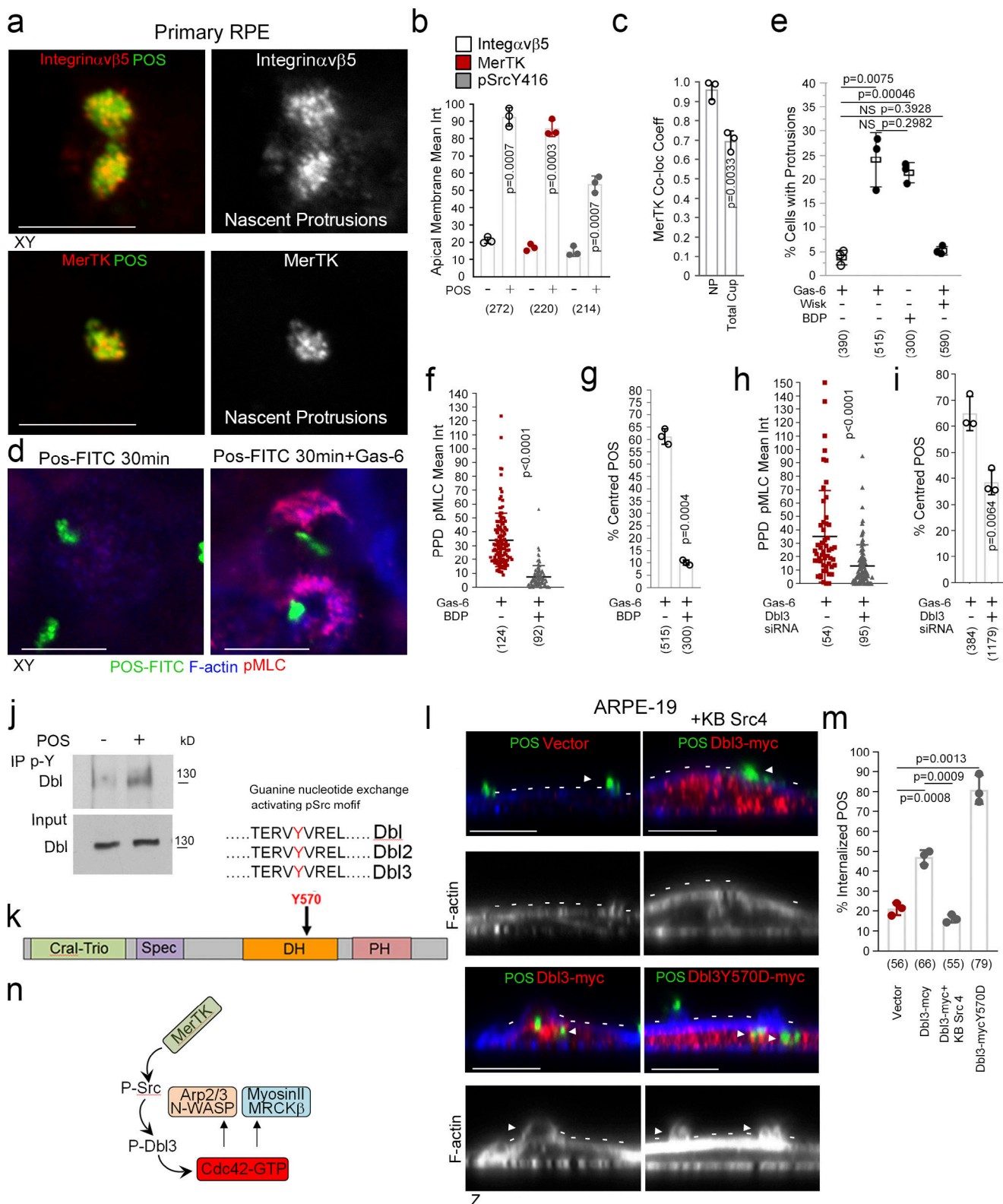

**Figure 8. MerTK stimulates Dbl3-activated actomyosin dynamics. (a and b)** Integrin αvβ5, MerTK tyrosine kinase receptor, and its effector Src are enriched at POS-induced nascent protrusions in primary porcine RPE cells. **(c)** Similar to integrin αvβ5, MerTK strongly colocalizes with POS at nascent protrusions. **(d–g)** Stimulating primary porcine RPE cells with POS in serum-deprived conditions does not strongly induce protrusions unless the ligand for MerTK, Gas-6, is added. Gas-6 promotes MRCK-dependent pMLC activity at cups and POS centering. **(h and i)** Dbl3 signaling is required for POS–Gas-6 stimulated pMLC activity at cups and POS centering. **(j)** Immunoblot showing immunoprecipitation (IP) of Dbl by anti-phosphotyrosine antibody following POS stimulation of primary porcine RPE cells. **(k)** Schematic illustration of a predicted Src/tyrosine phosphorylation motif in the Dbl3 DH domain. **(l and m)** Exogenous expression of Dbl3-myc wild type in the presence of a c-Src inhibitor blocks protrusion induction and POS engulfment in ARPE-19 cells,

whereas Dbl3Y570D-myc stimulates increased POS internalization. Arrowheads in F-actin images highlight protrusion and cup induction by Dbl3-myc. **(n)** Schematic illustration of proposed model in which phosphorylation of Dbl3 following MerTK stimulation of Src activates Cdc42 signaling. PH, pleckstrin homology. Scale bars represent 10 µm. All quantifications presented as bar plots are based on *n* = 3 independent experiments and show the data points, means ±1SD, whereas in f and h column scatter plots from three independent experiments highlight the median and upper and lower quartiles. *n* = the total number of cells analyzed for each type of sample across all experiments, and P values derived from *t* tests. Source data are available for this figure: SourceData F8.

guidelines on the care and use of animals adopted by the Society for Neuroscience and the Association for Research in Vision and Ophthalmology.

### Generation of CRISPR lentiviral particles and subretinal injection

Vesicular stomatitis virus G pseudotyped lentiviral particles were generated as described (Bainbridge et al., 2001). Subretinal injections were performed under direct retinoscopy through an operating microscope. The tip of a 1.5-cm, 34-gauge hypodermic needle (Hamilton) was inserted tangentially through the sclera of the mouse eye, causing a self-sealing wound tunnel. The needle tip was brought into focus between the retina and retinal pigment. Animals received double injections of 2 µl each to produce bullous retinal detachments in the superior and inferior hemispheres around the injection sites. Eyes were assigned as treated and (contralateral) control eyes using a randomization software.

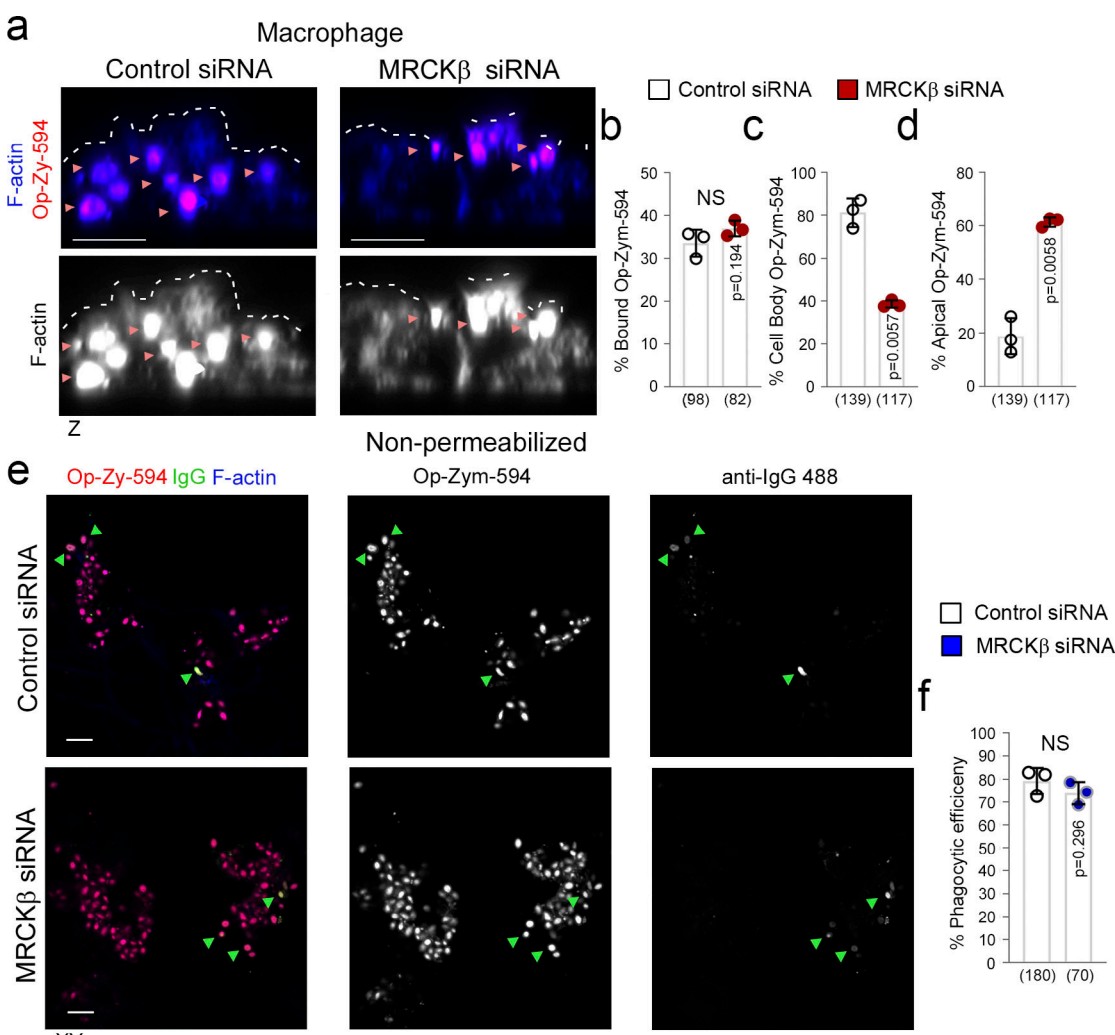

Figure 9. **MRCKβ is required late in FcR-macrophage phagocytosis. (a–d)** siRNA-mediated knockdown of MRCKβ in THP-1–derived macrophages demonstrates its requirement for FcR-mediated IgG-opsonized particle internalization into the cell body. Note that forming phagosomes are embedded in the cortex indicating particles may be tethered to the internal side of the membrane. **(e and f)** Nonpermeabilized cells were treated with anti–IgG-488 antibody to detect external bound opsonized Zymosan particles, when compared to total Zymosan-555 and to determine internalization efficiency. White dashed lines highlight the apically facing F-actin cortex, and green arrowheads highlight co-localization between Opsonized-Zymosan and IgG stains. Scale bars represent 10 µm. The quantifications are based on *n* = 3 independent experiments and show data points, means ±1SD, the total number of cells analyzed for each type of sample across all experiments, and P values derived from *t* tests.

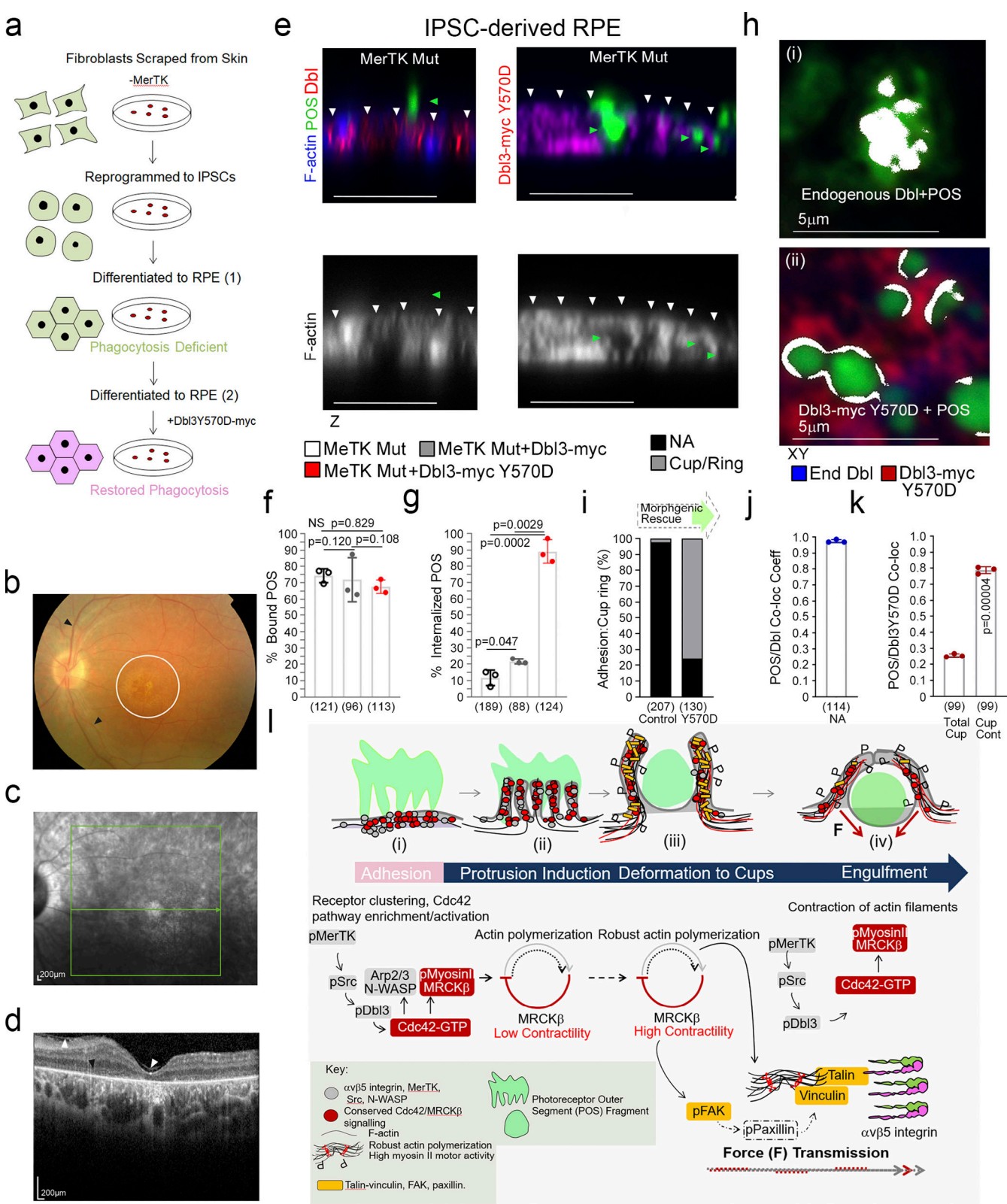

Figure 10. **Dbl3Y570D rescues phagocytosis in mutant MerTK iPSC-derived RPE cells. (a)** Schematic outline of RPE cell generation from the MerTK mutant fibroblasts. Note, fibroblasts were derived from a blind individual suffering from retinitis pigmentosa (RP). **(b)** Fundus image taken from a MerTK-deficient RP individual's right eye showing the retinal vessels (black arrowheads) and macula (white circle). **(c)** A red-free image of the macula with the green arrow representing the position of the line scan in b. **(d)** A spectral domain optical coherence tomography image of the left fundus, through the fovea, showing a thinning of the retina at the fovea (normally over 200 µm) and a loss of retinal lamination, especially centrally (total loss of ellipsoid zone, black arrowhead); the white arrowheads show an epiretinal membrane. **(e–g)** Confocal immunofluorescence z-section analysis of MerTK mutant RPE transfected with wild-type

Dbl3 or Dbl3Y570D and exposed to POS-FITC. Note, RPE cells expressing Dbl3Y570D undergo efficient recovery of phagocytosis. White arrowheads highlight the apical F-actin cortex. **(h–k)** Analysis of POS adhesion sites to cup population ratios in mutant cells and cells expressing Dbl3Y570D. Note, colocalization analysis by Pearson's coefficient calculation reveals that MerTK-deficient RPE cells contain almost exclusively Dbl-rich adhesion sites that fail to transform into protrusions depicted in white, whereas mutant cells expressing Dbl3Y570D undergo cup maturation with characteristic Dbl-POS colocalization at contact sites depicted as white rings at contact points. Scale bars represent 10 μm, unless highlighted otherwise. All quantifications are based on $n = 3$ independent experiments and show the data points, means ±1SD, the total number of cells analyzed for each type of sample across all experiments, and P values derived from $t$ tests are indicated. **(l)** Schematic illustration of proposed model of MRCKβ-controlled morphodynamic signaling in MerTK receptor-mediated RPE cells. Briefly, MerTK-Cdc42–driven MRCKβ signaling progresses through distinct phases with a lower activity during initial pseudopod induction and high activity, concomitant with increased F-actin, during deformation of protrusions to mediate particle wrapping. Continued MRCKβ-driven myosin-II activity stimulates FAK activation of paxillin (dashed outline represents a putative link based on the established mechanosensory bridge in migratory cells), recruitment of mechanotransducers vinculin and talin, and receptor clustering at cups to coordinate particle wrapping.

## Mammalian antibodies and immunological methods

Fixation and processing of cells and mouse tissue sections was as previously described (Zihni et al., 2017). The following antibodies were used: RPE65 mouse monoclonal (Millipore) 1/300 for immuno-fluorescence; p-MLC S19, mouse monoclonal (Cell Signaling Technology) immunofluorescence 1/100 and immunoblotting 1/1,000; Dbl(3), rabbit polyclonal (Santa Cruz Biotechnology) immunofluorescence 1/200; MRCKβ, rabbit polyclonal (Santa Cruz Biotechnology) immunofluorescence 1/200 and immunoblotting 1/500; anti–Cdc42-GTP mouse monoclonal (NewEast Biosciences) immunofluorescence 1/50; N-WASP, rabbit polyclonal (Santa Cruz Biotechnology) immunofluorescence 1/200 and immunoblotting 1/1,000; MerTK rabbit monoclonal (52968; Abcam; recognizes an extracellular N-terminal epitope) immunofluorescence 1/100; Integrinα$_v$β$_5$, mouse monoclonal (P1F6; Clone; 24694; Abcam) immunofluorescence 1/100; pFAK Y397, rabbit polyclonal (Invitrogen) immunofluorescence 1/100; pSrc Y416, rabbit polyclonal (Cell Signaling Technology) immunofluorescence 1/100; Phalloidin-Atto 647 reagent was obtained from Sigma-Aldrich and diluted 1/1,000. Affinity-purified and cross-adsorbed Alexa488-, Cy3- and Cy5-labeled donkey anti-mouse, rabbit, or goat secondary antibodies were from Jackson ImmunoResearch Laboratories (1/300 diluted from 50% glycerol stocks). Affinity-purified HRP-conjugated goat anti-mouse and rabbit, and donkey anti-goat secondary antibodies 1/5,000 were also from Jackson ImmunoResearch Laboratories (1/5,000 diluted from 50% glycerol stocks). For immunofluorescence analysis, cells and tissues were mounted using Prolong Gold antifade reagent (Life Technologies), and imaging was performed using Zeiss 700 and 710 confocal microscopes and a 64× oil lens/NA1.4. Images were processed using Zeiss Zen2009 and Adobe Photoshop CS5 and 10 software. Coimmunoprecipitation and immunoblotting were carried out using methods previously described and were repeated at least three times (Zihni et al., 2014; Zihni et al., 2017).

## Isolation of RPE and POS

Pig's eyes were obtained from a slaughterhouse. Briefly, in chilled and sterile conditions, each eye was cut into two halves (cornea, including 5 mm into the sclera). After removing the lens, the eyeball was filled with PBS and the neural retina was removed using forceps. The PBS was then replaced with 10× trypsin and incubated at 37°C for 30 min. The trypsin was pipetted several times to dislodge the RPE cells and centrifuged at RT at 800 rpm for 5 min and washed with PBS. The cells were plated into 6-well tissue culture plates with DMEM containing 10% FBS and antibiotics. To isolate POS, pig's eyes were treated as for RPE isolation, except that following removal of the lens the neural retina was scraped off the tapetum surface and collected in tubes containing homogenization solution. The solution was vigorously shaken for 2 min and passed through a double-layer gauze to remove tissue fragments. The crude retina preparation was poured into a 30-ml ultracentrifuge tube containing chilled continuous sucrose gradients (25–60%) and centrifuged in a Beckman SW-27/SW-32-Ti swing rotor at 25,000 rpm at 4°C for 1 h. The appropriate layer was collected and washed three times with centrifugation at 5,000 rpm for 10 min each time. The samples were resuspended in DMEM and stored at –80°C. Isolated POS was labeled before use by centrifuging samples at 6,300 rpm at RT for 10 min and washing before sonicating for 10 min. FITC label was added to the POS and incubated for 2 h whilst rotating 4°C. The samples were then washed 5× in PBS and suspended in appropriate medium.

## EM

All steps were performed at RT as previously described (Zihni et al., 2014; Zihni et al., 2017). Briefly, cell monolayers were fixed in a mixture of 3% (vol/vol) glutaraldehyde and 1% (wt/vol) PFA in 0.08 M sodium cacodylate buffer, pH 7.4, for 2 h at RT and left overnight at 4°C. Before osmication, the primary fixative solution was replaced by a 0.08 M cacodylate buffered solution of 2.5% glutaraldehyde and 0.5% (wt/vol) tannic acid. After two brief rinses in cacodylate buffer, specimens were osmicated for 2 h in 1% (wt/vol) aqueous osmium tetroxide, dehydrated by 10 min incubations in 50, 70, 90, and three times in 100% ethanol. Semithin sections (0.75 μm) for light microscopy and ultrathin sections (50–70 nm) for EM were cut from sawed-out blocks with diamond knives (Diatom; Leica). Semithin sections were stained with 1% toluidine blue/borax mixture at 60°C and ultrathin sections were stained with Reynold's lead citrate. Stained ultrathin sections were examined in a TEM (1,010; JEOL) operating at 80 kV, and the images were recorded using an Orius B digital camera and Digital Micrograph (Gatan, Inc.)

## Statistical analysis

POS internalized into the cell body were measured in Zeiss 700 confocal z-sections by analyzing FITC-labeled POS particles using Zeiss Zen2000 imaging software or by analyzing Gold-labeled POS by TEM. The percentage of cells with attached POS-FITC was quantified using a confocal $xy$ section ranging from the apical cortex to the basal cortex (using phalloidin-Cy5 as

a cortex marker). Cells that contained either externally attached POS-FITC and/or internally attached POS-FITC at ≥1 were scored positive. Alternatively, nonpermeabilized cells were labeled with anti-annexin V antibody. As the neural retina is rich in annexin V (Yu et al., 2019), its homogenization during POS isolation results in labeling of POS due to the affinity of annexin V for phosphatidylserine, which is exposed by POS. Colocalization of annexin V with POS-FITC represents external POS, which is subtracted from the total number of POS-FITC particles to determine internalized POS and then divided by total POS to calculate phagocytosis efficiency. Cdc42 activation was analyzed using a commercially available monoclonal antibody that specifically recognizes Cdc42 in its GTP-bound form. We have previously reported data supporting the specificity of this antibody, demonstrating that it only stains Cdc42 in immunofluorescence assays upon activation by Cdc42 exchange factors (Elbediwy et al., 2012). For in vivo analysis, microvillar organization was measured in electron micrographs using ImageJ software. Sheets of microvilli comprising 0, 0–5, or >5 width of microvilli at nm intervals were measured as units every 2.5 μm image width. Internalized phagosomes and photoreceptor outer segment fragmentation were measured in electron micrographs using Zeiss Zen2000 software in ~20 μm fields. External POS fragments were quantified as the total number in contact with the RPE apical membrane/20 μm as a sample value of the total neural retinal. ONL thickness, outer limiting membrane intensity, and ratio of apical:basal F-actin cortex intensity were measured using confocal immunofluorescence sections at ~200-μm widths of retinal tissue. Immunostaining pixel intensity was measured using ImageJ software. For each measurement, the background was measured and subtracted from the sample value. The control datasets for the immunofluorescence experiments Fig. S4, f and g, are the same dataset as controls in Fig. 7 d (as Control, MRCKβ 1, and MRCKβ 2 were carried out as part of the same independent sets of experiments). For localization of signaling proteins at nascent protrusions, xy immunofluorescence confocal sections were measured at the apical membrane of POS-stimulated cells both at areas where POS was absent and attached in the same population as an internal control (CF1). Where nascent membrane protrusions were higher than normal microvilli protrusions, the membrane was scanned an appropriate number of equal serial sections. For scatter plots of nascent protrusion, the individual data points of αvβ5 integrin, pMLC, vinculin, and talin were generated by subtracting first background and then regions of the apical membrane that were not attached to POS from attached regions, in the same population as an added internal control. Therefore, graphical Y-axis 0 values = labeling intensity of apical membrane devoid of POS attachment. Colocalization coefficients were measured in Zeiss confocal XY scans using Zeiss Zen2000 colocalization software that applies the principle of Pearson's coefficient to select subcellular structures and uses cross hairs function to subtract background from each channel. Colocalization coefficients at nascent protrusions, total cups, or POS-signaling plaque contacts at cups were measured by selecting these areas, indicated by colored outlines. The control datasets for Fig. 4, q–v, and Fig. S1, a and b, are the same datasets as controls for Fig. 5, j–n, and Fig. S3, a and b, as they were carried out as part of the same independent sets of experiments. For the

quantifications shown, the provided n values refer to independent experiments, and the numbers added in brackets in the graphs refer to the total of analyzed cells or number of field widths in electron micrographs and confocal immunofluorescence of mouse retinal tissue. Statistical significance was tested in experiments using two-tailed Student's t tests or ANOVA tests with an n value of at least 3.

### Online supplemental material
Fig. S1 contains additional data on Cdc42-driven protrusion formation upon POS binding to RPE cells. Fig. S2 shows that POS binding induces membrane deformation. Fig. S3 contains additional data on the role of MRCKβ in cup morphogenesis and particle internalization in RPE cells. Fig. S4 contains additional data on the importance of MRCKβ for retinal integrity in vivo. Fig. S5 contains additional data on the analysis of MerTK signaling during phagocytosis.

### Acknowledgments
This work was supported by grants from the Biotechnology and Biological Sciences Research Council (BB/N014855/1 and BB/N001133/1), Moorfields Eye Charity, and the Rosetrees Trust to K. Matter, M.S. Balda, and C. Zihni; Retina UK, National Institute for Health and Care Research Biomedical Research Centre at Moorfields Eye Hospital National Health Service Foundation Trust support to R.R. Ali; and the UCL Institute of Ophthalmology. C. Zihni is supported by a Moorfields Eye Charity Personal Fellowship.

Work reported here has been filed for patent protection by UCL Business (PCT/GB2021/050402). The authors declare no competing financial interests.

Author contributions: C. Zihni performed the majority of the experiments. All other authors performed particular subsets of experiments or provided technical support (E. Sanchez-Heras). A. Georgiadis, O. Semenyuk, J.W.B. Bainbridge, A.J. Smith, and R.R. Ali generated the lentiviral particles, performed subretinal injections, and participated in the analysis of the mouse eyes. C.M. Ramsden, B. Nommiste, P.J. Coffey, C. Zihni, and M.S. Balda generated the iPSC lines and differentiated the RPE cells. A.J. Haas supported the data quantification and analysis. C. Zihni, M.S. Balda, and K. Matter designed the project, planned the experimental approach, and drafted the manuscript. All authors read and contributed to the final version of the manuscript.

Submitted: 7 December 2020

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

# Supplemental material

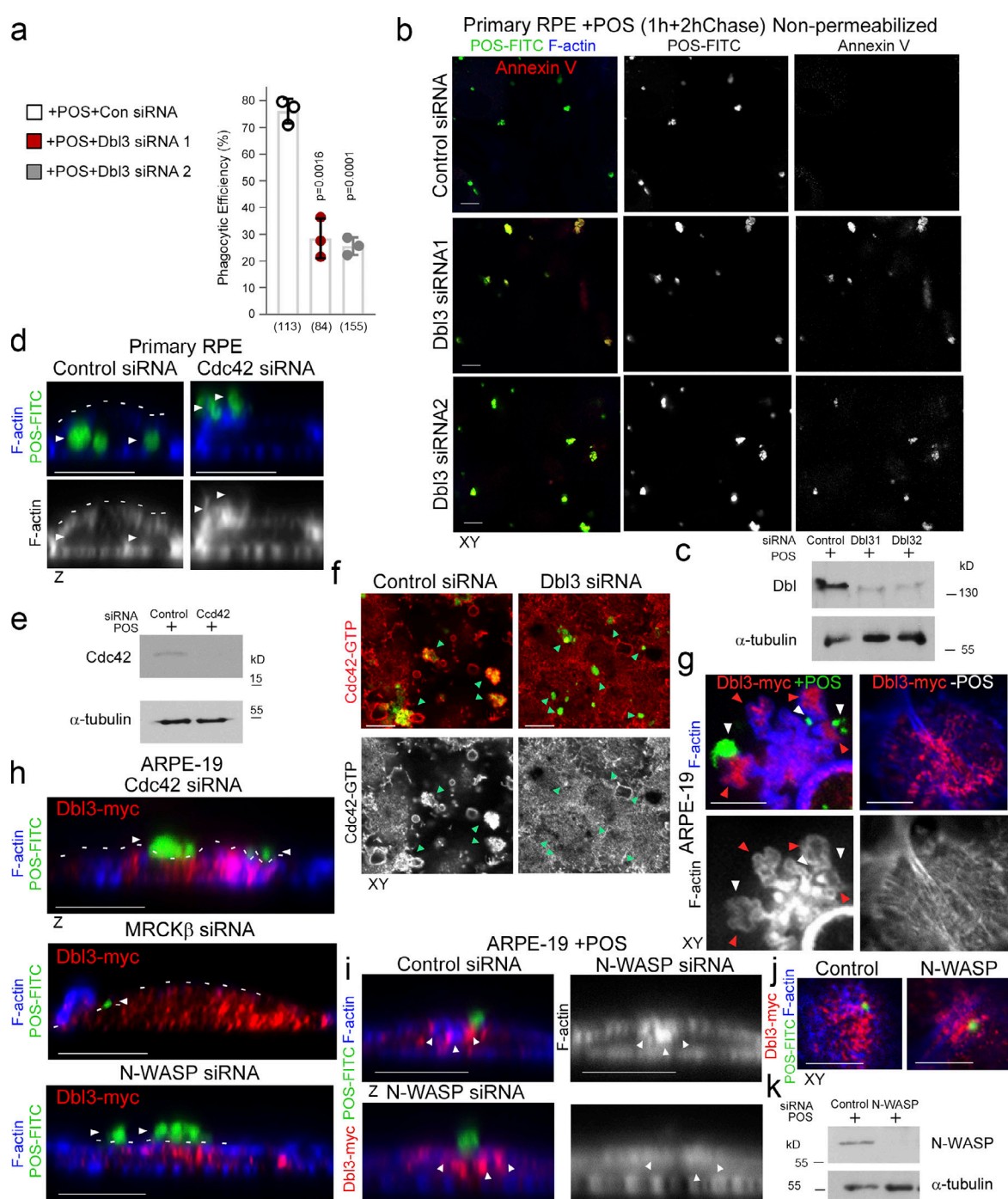

Figure S1. **Cdc42 drives protrusion induction and cup maturation. (a–c)** Confocal *xy* scanning analysis of Dbl3-depleted cells using two distinct Dbl3 siRNAs in primary porcine RPE cultures treated with POS-FITC for 1 h followed by 2 h chase. Cells were fixed and not permeabilized to detect external bound POS using anti-Annexin 5 antibody, in addition to total POS-FITC. **(d and e)** siRNA-mediated knockdown of Cdc42 in primary porcine RPE cultures treated with POS-FITC for 1 h followed by 2 h chase inhibits phagocytosis as determined by confocal *z*-section analysis of porcine primary RPE cells. White arrowheads highlight the position of POS, and white dashed lines highlight the apical F-actin cortex. **(f)** Active Cdc42-GTP at nascent protrusions is inhibited by siRNA-mediated knockdown of Dbl3. Note, green arrowheads highlight POS-membrane bound sites. **(g)** Confocal *xy* scanning analysis of ARPE-19 cells expressing Dbl3-myc with or without POS stimulation. Note, POS adhesion stimulates Dbl3-driven actin-rich cup formation but Dbl3-myc exogenous expression only does not stimulate cups. White arrowheads indicate cups attached to POS. Red arrowheads highlight Dbl3-myc localization at cup tips. **(h)** Confocal z-scanning analysis showing effects of siRNA-mediated knockdown of Cdc42 and its effectors in Dbl3-myc expressing ARPE-19 cells. Note, knockdown of Cdc42 signaling in Dbl3-myc–driven phagocytosis results in an inhibition of phagocytosis. White arrowheads highlight the position of POS, and white dashed lines highlight the apical F-actin cortex. **(i–k)** Confocal xy and z sections of ARPE-19 cells with N-WASP knockdown expressing Dbl3-myc and stimulated with POS-FITC for 30 min reveals cells require N-WASP for F-actin polymerization during protrusion induction. White arrowheads highlight loss of Dbl3-myc induced F-actin staining revealed by *z*-section analysis at sites of POS attachment following N-WASP siRNA-mediated knockdown. Analyzes in a is based on *n* = 3 independent experiments. Source data are available for this figure: SourceData FS1.

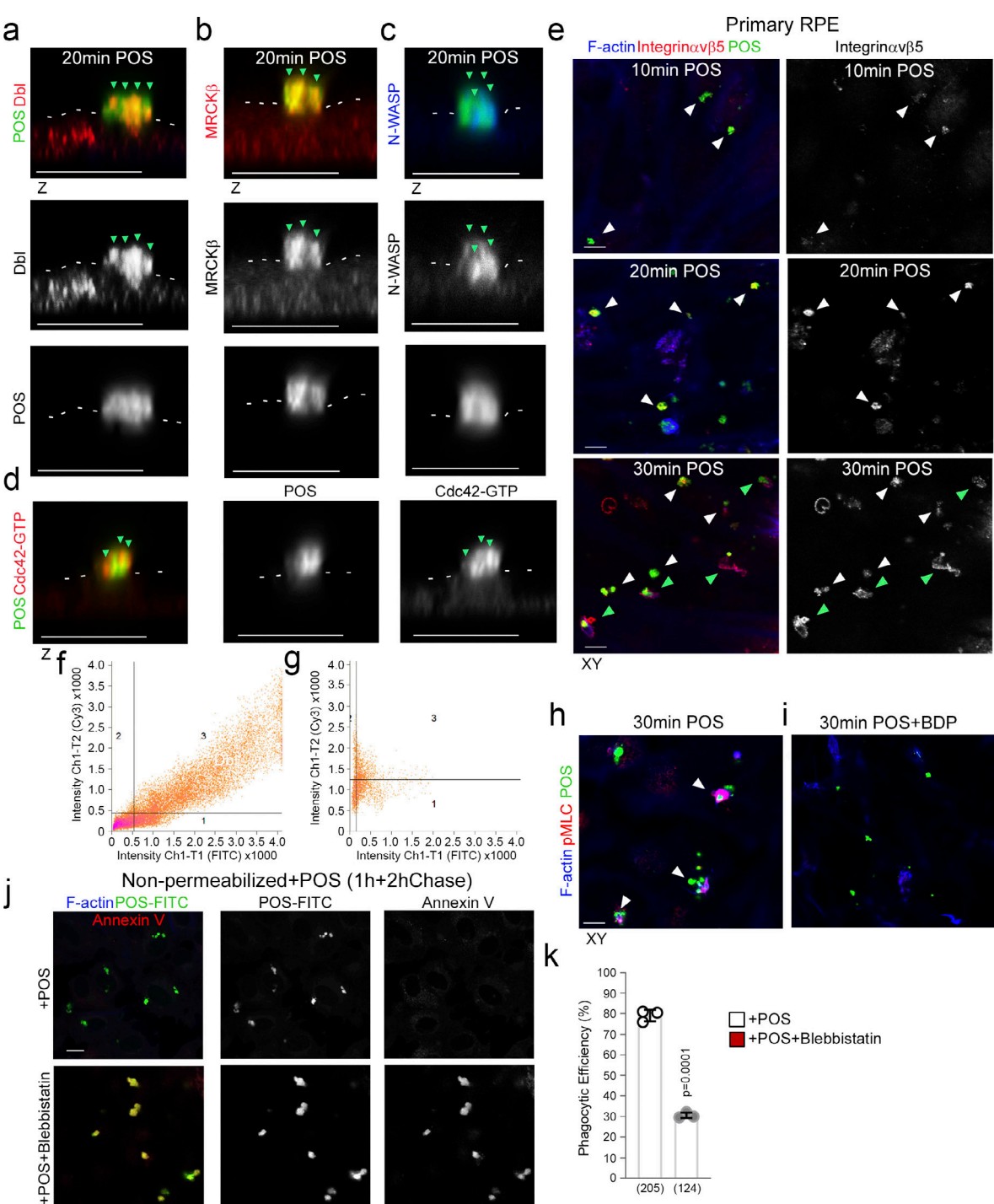

Figure S2. **POS-membrane contacts induce membrane deformation. (a–d)** Confocal *z*-section analysis reveals that POS binding to the apical membrane of RPE, induces membrane protrusions rich in Dbl3, Cdc42-GTP, MRCKβ, and N-WASP. White dashes highlight the normal apical membrane. **(e)** Time course of POS incubation with primary RPE cells reveals that adhesion sites are enriched with receptor within 10 min of POS addition, as determined by adhesion receptor αvβ5 integrin staining and peak with respect to mean labeling intensity at 20 min. At 20 min, membrane deformation results in protrusions that transform into cups that wrap around POS at 30 min. Note, white arrowheads highlight POS-membrane adhesion sites and green arrowheads highlight cups. **(f and g)** Primary porcine RPE cells were treated with POS-FITC for 20 or 30 min and analyzed by confocal microscopy using colocalization software based on the Pearson's correlation coefficient. Colocalization was calculated in grid three of the scatter charts. **(h and i)** Confocal xy scans of porcine primary RPE cells treated with POS for 30 min reveal cups encircling the particles with increased pMLC activity compared to POS-free segments of apical membrane. Treatment of cells with the MRCK inhibitor BDP5290 results in pseudopods negative for pMLC that are morphogenetically defective and unable to wrap around POS. White arrowheads highlight phagocytic cups. **(j and k)** Confocal xy scanning analysis of blebbistatin-treated primary porcine RPE cultures treated with POS-FITC for 1 h followed by 2 h chase. Cells were fixed and not permeabilized to detect external bound POS using anti-Annexin 5 antibody, in addition to total POS-FITC. Protein staining was quantified as mean intensity. Scale bars represent 10 μm. Analysis in k is based on *n* = 3 independent experiments and shows the data points, means ±1SD, the total number of cells analyzed for each type of sample across all experiments, and P values derived from *t* tests.

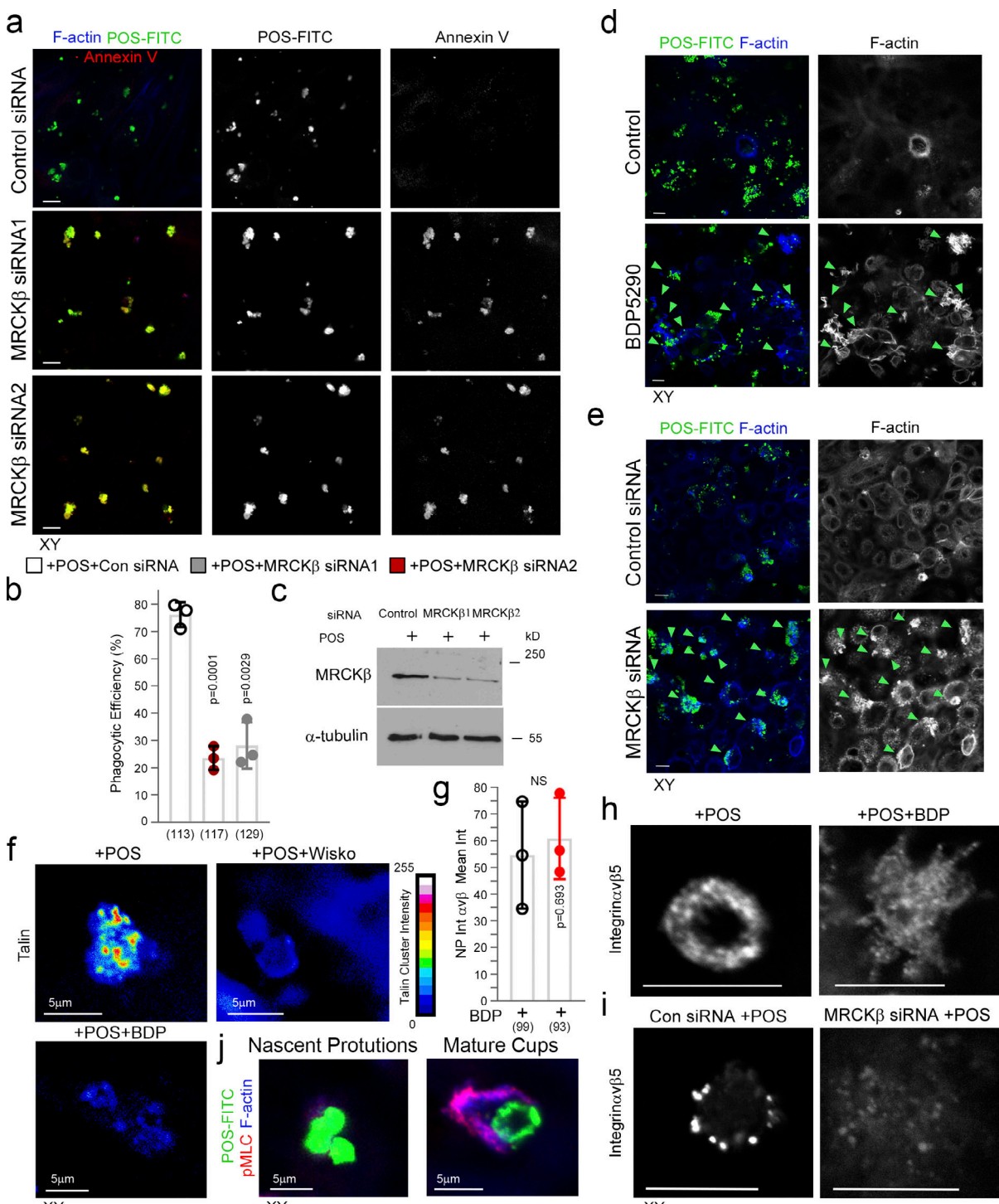

Figure S3.   **MRCKβ is required for cup morphogenesis and particle internalization. (a–c)** Confocal xy scanning analysis of Dbl3-depleted cells using two distinct MRCKβ siRNAs in primary porcine RPE cultures treated with POS-FITC for 1 h followed by 2-h chase. Cells were fixed and not permeabilized to detect external bound POS using anti–Annexin 5 antibody in addition to total POS-FITC. **(d)** siRNA-mediated knockdown of MRCKβ in primary porcine RPE cells leads to prolonged, longer, and disordered pseudopods with increased F-actin labeling. **(e)** Inhibition of MRCKβ kinase activity using BDP5290 results in a membrane remodeling defect in a similar manner to knock-down of MRCKβ. Note, POS is often misaligned and not centered at extended pseudopods in either MRCKβ siRNA-treated or BDP5290-treated cells, as indicated by green arrowheads in the confocal xy-scanning sections. **(f)** N-WASP and MRCK inhibitors in cells treated with POS for 30-min block clustering of mechanosensory talin. **(g–i)** Confocal xy analysis of integrin αvβ5 distribution in cups with or without MRCKβ inhibition or knockdown. Note, inhibition of MRCKβ-driven actomyosin contractility results in loss of integrin αvβ5 clustering along the deformed pseudopods that do not mature into normal cylindrical cups, as highlighted by F-actin staining. **(j)** pMLC and F-actin labeling increase during transition from nascent protrusion to cup formation. Scale bars represent 10 μm. All quantifications are based on n = 3 independent experiments and show the data points, means ±1SD, the total number of cells analyzed for each type of sample across all experiments, and P values derived from t tests. Source data are available for this figure: SourceData FS3.

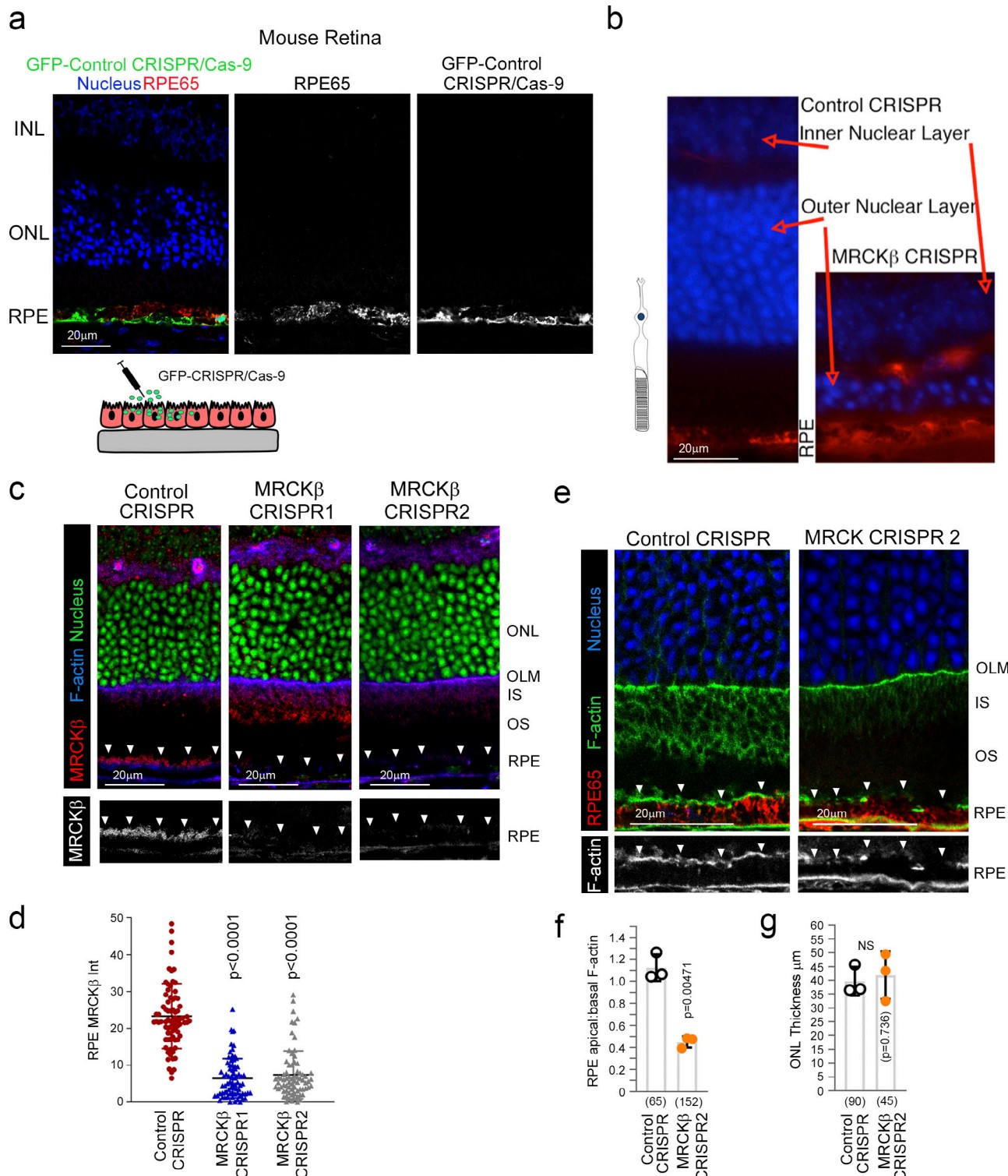

Figure S4. **MRCKβ is required for retinal integrity and function in vivo. (a)** Subretinally injected lentiviral vectors specifically infect the RPE. Shown is GFP expression upon transduction with a control vector, combined with staining for the RPE marker RPE65 and DNA. **(b)** Frozen sections of mouse eyes 21 d after injection with either control or MRCKβ-targeting viruses were stained for RPE65 and DNA. Note, MRCKβ knockout leads to a pronounced retinal degeneration as determined by extensive thinning of the ONL. **(c and d)** MRCKβ expression upon transduction with control CRISPR or two distinct viruses targeting the MRCKβ gene. Note, apical membrane expression, highlighted by white arrowheads, of MRCKβ observed in control tissue samples is lost in both CRISPR MRCKβ 1 and 2. **(e and f)** Confocal immunofluorescence analysis of retinal sections from mice injected with either control or MRCKβ 2 knockout vector; as with MRCKβ 1 knockout vector, no significant difference in the overall retinal structure after 7 d but reduced apical F-actin staining, with reference to basal F-actin, in the MRCKβ-deficient RPE. INL, inner nuclear layer; IS, inner segments; OS, outer segments. White arrows highlight apical F-actin cortex. Protein staining was quantified as mean intensity (Int). Quantifications represented by bar graphs show means ± 1SD, n = 3, and P values derived from t tests. In d, a column scatter plot from three independent experiments shows the median and upper and lower quartiles. P values are derived from ANOVA tests.

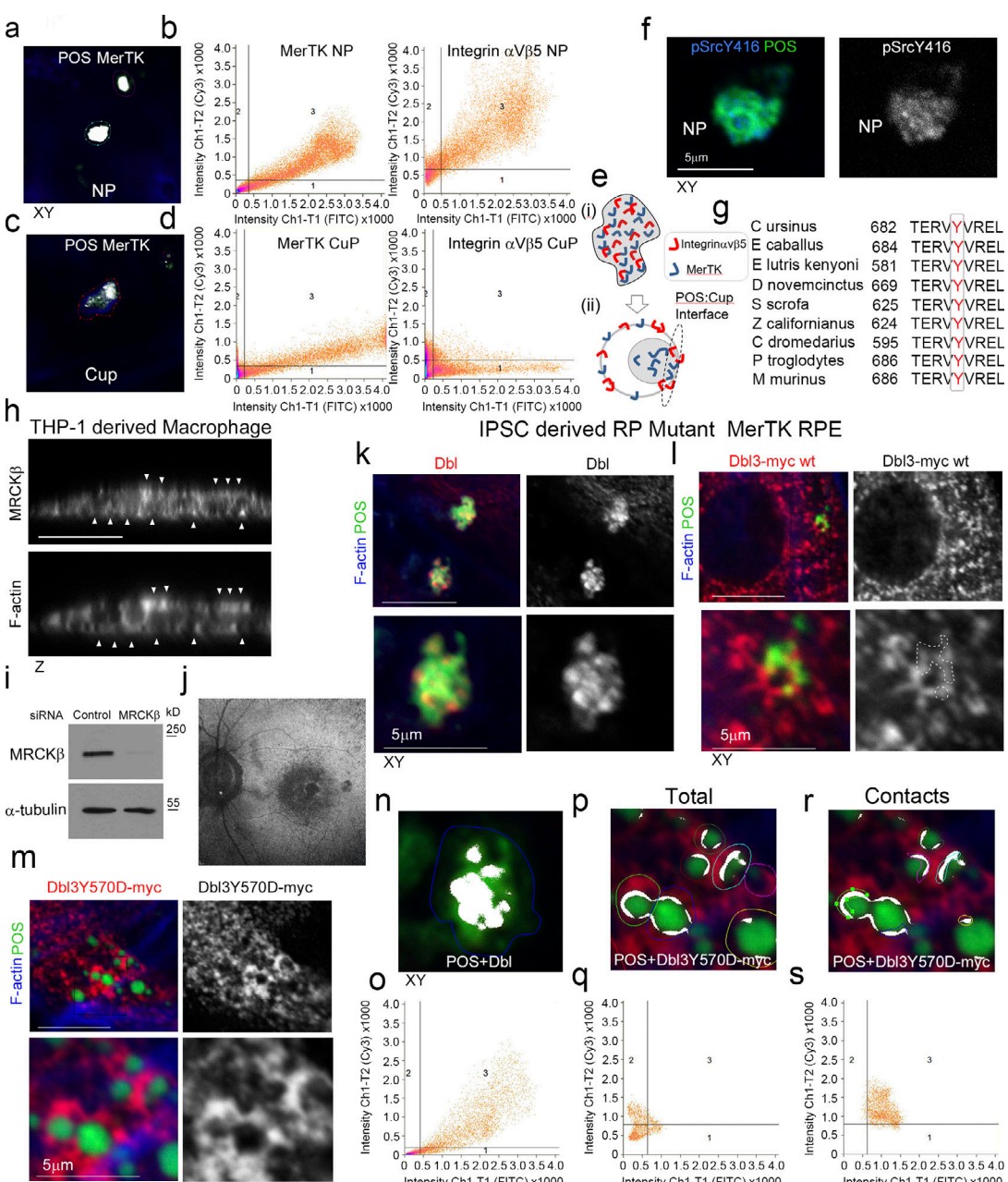

Figure S5. **MerTK-signaling during phagocytosis. (a–e)** Primary porcine RPE cells were analyzed by confocal xy scanning microscopy for localization of the receptor relative to POS using colocalization software based on the Pearson's correlation coefficient. Grid 3 of the scatter charts contains the colocalization data calculations. Colocalization of POS-FITC with MerTK is depicted in white. Note, colocalization of MerTK with membrane-bound POS is nearly total at nascent protrusions (NP). MerTK also localizes at cups and contact sites but more substantially with total POS. Colored outlines of nascent protrusions, cups, or POS-cup contacts represent areas used for calculating of colocalization coefficients. **(f)** Confocal microscopy analysis of primary RPE treated with POS-FITC revealed activated MerTK receptor-associated tyrosine kinase Src (pSrcY416) at nascent protrusions. **(g)** Sequence alignment of the Dbl GEF motif for activation by c-Src tyrosine phosphorylation in diverse mammalian species demonstrating high evolutionary conservation. **(h)** Confocal z-sectional analysis of THP-1–derived macrophages reveal MRCKβ localizes along the actin cortex. White arrowheads highlight apical- and basal-facing sections of the F-actin cortex. **(i)** Immunoblot demonstrating siRNA knockdown of MRCKβ in THP-1–derived macrophages. **(j)** Autofluorescence image of a magnified area of the posterior pole. The speckled darker area at the central macula represents a dropout of RPE cells from an individual carrying a loss of function MerTK mutation that causes retinitis pigmentosa (RP) and vision loss. **(k)** confocal xy scanning fluorescence image of RPE cells differentiated from iPSCs derived from an RP individual reveals an accumulation of adhesion sites with enriched Dbl at POS contact sites but no wrapping of POS. **(l and m)** Expression of Dbl3Y570D-myc tyrosine phospho-mimetic mutant promotes efficient nascent adhesion maturation despite loss of upstream MerTK function. **(n–s)** RP individual iPSC-derived RPE expressing different forms of Dbl analyzed by confocal xy scanning microscopy using colocalization software based on the Pearson's correlation coefficient. Note, colored outlines in the confocal xy-section images represent areas of calculation. Dbl3Y570D fully restores progression from almost complete colocalization at adhesion sites to POS-membrane Dbl3 interface contacts. Note, Grid 3 of the scatter graph represents colocalization data calculations and Dbl colocalization with POS is highlighted in white. All quantifications are based on n = 3 independent experiments. Source data are available for this figure: SourceData FS5.

