## [Peer Review File · The Journal of Cell Biology]

Spatiotemporal Control of Actomyosin Contractility by MRCKbeta Signaling Drives Phagocytosis

Ceniz Zihni, Anastasios Georgiadis, Conor Ramsden, Elena Sanchez-Heras, Alexis Haas, Britta Nommiste, Olha Semenyuk, James Bainbridge, Peter Coffey, Alexander Smith, Robin Ali, Maria Balda, and Karl Matter

Corresponding Author(s): Karl Matter, University College London and Ceniz Zihni, University College London

Review Timeline:

Submission Date:	2020-12-07
Editorial Decision:	2021-01-19
Revision Received:	2022-07-08
Editorial Decision:	2022-07-25
Revision Received:	2022-08-17

Monitoring Editor: Xiaochen Wang

Scientific Editor: Andrea Marat

Transaction Report:

DOI: <https://doi.org/10.1083/jcb.202012042>

January 19, 2021

Re: JCB manuscript #202012042

Prof. Karl Matter
University College London
Division of Cell Biology UCL Institute of Ophthalmology University College London
Bath Street
London EC1V 9EL
United Kingdom

Dear Prof. Matter,

Thank you for submitting your manuscript entitled "Spatiotemporal Control of Actomyosin Contractility by MRCKbeta Signaling Drives Phagocytosis" and thank you for your patience with the peer review process. The manuscript was assessed by expert reviewers, whose comments are appended to this letter. We invite you to submit a revision if you can address the reviewers' key concerns, as outlined here.

You will see that a major consistent issue is the insufficient evidence for your model implicating myosin contractility in phagocytosis. We have discussed the reviewer feedback editorially and find the reviewers' comments valid, with addressable and constructive suggestions. The role of myosin II and myosin contractility should be clearly tested and demonstrated (concerns raised by both Revs#2 and #3), given that this has been controversial in the field and is the major point of the manuscript. In this regard, the specific points regarding the role of myosin II and myosin contractility should be addressed. Please also address the point regarding phagocytic efficiency raised by Rev#3. This will help to clarify the role of myosin in phagocytosis in RPE and macrophages. The suggested experiments are also practical. The MRCKb experiments suggested by Rev#1, including additional ways to enhance actin depolymerization, should be performed to firmly support the conclusions. The control experiments suggested by Rev#2 should be included. Please let us know if you have any questions or anticipate any issues addressing the reviews. We would be happy to further discuss as needed.

GENERAL GUIDELINES:

Text limits: Character count for an Article is < 40,000, not including spaces. Count includes title page, abstract, introduction, results, discussion, acknowledgments, and figure legends. Count does not include materials and methods, references, tables, or supplemental legends.

Figures: Articles may have up to 10 main text figures. Figures must be prepared according to the policies outlined in our Instructions to Authors, under Data Presentation, <https://jcb.rupress.org/site/misc/ifora.xhtml>. All figures in accepted manuscripts will be screened prior to publication.

Supplemental information: There are strict limits on the allowable amount of supplemental data. Articles may have up to 5 supplemental figures. Up to 10 supplemental videos or flash animations are allowed. A summary of all supplemental material should appear at the end of the Materials and methods section.

As you may know, the typical timeframe for revisions is three to four months. However, we at JCB realize that the implementation of social distancing and shelter in place measures that limit spread of COVID-19 also pose challenges to scientific researchers. Lab closures especially are preventing scientists from conducting experiments to further their research. Therefore, JCB has waived the revision time limit. We recommend that you reach out to the editors once your lab has reopened to decide on an appropriate time frame for resubmission. Please note that papers are generally considered through only one revision cycle, so any revised manuscript will likely be either accepted or rejected.

We hope that the comments below will prove constructive as your work progresses. We would be happy to discuss them further

once you've had a chance to consider the points raised in this letter.

Thank you for this interesting contribution to Journal of Cell Biology. You can contact us at the journal office with any questions, cellbio@rockefeller.edu or call (212) 327-8588.

Sincerely,

Xiaochen Wang, PhD
Monitoring Editor, Journal of Cell Biology

Melina Casadio, PhD
Senior Scientific Editor, Journal of Cell Biology

Reviewer #1 (Comments to the Authors (Required)):

In the study by Zihni et al., they mainly deployed the retinal pigment epithelium (RPE), supplemented with CRISPR-Cas9 modified mice, macrophages, and iPSC, to investigate the role of the MerTK signaling in the various stages of phagocytosis. Notably, they found that myosin-II activator MRCKb is required for regulation of actomyosin contractility to drive phagocytosis. In this process, receptor MerTK activates the Cdc42-GEF Dbl3 through Src-mediated phosphorylation. Then, active Cdc42 stimulates N-WASP and MRCKb. N-WASP promotes actin-based pseudopod-like protrusions, while MRCKb activates myosin-II to limit actin assembly and induces deformation of pseudopods into cups. These data suggested local regulatory feedback between N-WASP-assisted actin polymerization and MRCKb-stimulated myosin-II activation, actin disassembly, and contractility. Additionally, they proposed that MRCKb-driven contractility also induces the assembly of a mechanosensory bridge consisting of activated FAK, talin, and vinculin, which promotes integrin clustering to guide membrane wrapping POS. In addition to the RPE system, they also tested the mechanistic conservatism of MerTK signaling during FcR-mediated macrophage phagocytosis and animals retinal integrity in vivo. Finally, through using a phosphomimetic Dbl3 mutant, they demonstrated that Dbl3-signaling is sufficient to bypass MerTK to induce phagocytosis in iPSC-derived retinitis pigmentosa RPE cells, highlighting that the major role of MerTK receptor is to activate MRCKb-signaling.

The findings of this study provide a nice addition to understanding how the phagocytosis is spatiotemporally regulated. In particular, it elaborates and consolidates the functional significance of the Cdc42-signaling in phagocytosis, which has been proposed in the field. I have some criticisms, which could help clarify some confusion and help understand the functional model presented in this manuscript.

Major comments:

Page 3, Cdc42 has been involved in regulating the formation and stability of cell junctions, primarily through actin and vesicular trafficking. In addition to the model proposed, does the activation of Cdc42 promote the directional transport of integrin and thus assist clustering?

Is there a time point choice for rescue experiments in iPSC, such as different stages of phagocytosis?

Page 5, "These results support a role for MRCKb driven contractility as a regulator of actin dynamics during protrusion maturation into cups." The MLC phosphorylation level should be measured after the knockdown of MRCKb to reach this conclusion. Please adjust the language.

Page 5, "MRCK activity is not required for pseudopod induction but for remodeling of protrusions to form cups, particle wrapping, and internalization." The formation mechanism of pseudopod should be carefully discussed and preliminarily tested, which is very important for refining this research's conceptual advance. Also, this can help clarify the critical regulatory modes in the phagocytosis.

Page 6, Fig 2u. This model is very interesting, but there is insufficient evidence to sustain this model directly. For example, knockdown of MRCK and Myosin II, or overexpression of MRCK and Myosin II should be used in cells overexpressing N-WASP to detect whether the phenotype is enhanced or weakened.

Fig 3, in addition to the inactivation of MRCKb, other methods could be used to enhance the actin depolymerization and see if it has the same effect, thus verifying this model.

Fig 3, does the use of inhibitors reduce the recruitment of clutch molecules? Please show.

Fig 3p, the data to support this model is insufficient. For example, it is recommended to display the colocalization of clutch molecules with MRCK and/or integrin at different stages, as well as the F-actin level changes at the same time scale.

In Fig 5g left pathway panel, there is an arrow pointing from MRCK to N-WASP, is there any evidence supporting this claim?

Given the model proposed in Fig 5g, it is recommended that Fig 4 and Fig 5 be moved to the end, and Fig 6 and Fig 7 are moved after Fig 3 to keep the story smooth.

Minor comments:

Fig 1, Page 4, there is a lack of explanation for the labels, and the Y-axis label of the statistical graph is covered. It is recommended to add an RPE diagram for explanation purposes.

Page 4, POS binding assay, please explain it briefly. Also, please explain how to assay Cdc42 activation.

Figure 1i, red arrows are hard to find.

The POS label in Fig 1j is unclear. Please indicate.

It is recommended to give a schematic diagram of the early and late events, adding into Fig. 3, which helps readers understand the significance and importance of this part of data.

Fig 3, EM shows that the cup cannot be observed after the BDP5290 application. Is this the reason for the lack of phagocytosis? The statistical information of c, f, h is incomplete, and no specific statistical indicators can be seen.

Fig 4, j matches l, but i and k do not match; please verify this. Also, please indicate where the h was cropped out. I did not see phagosomes in MRCK-deleted cells.

Viculin is a typo, should be vinculin.

In general, the lack of annotations for many symbols in figure legends leads to difficulties in reading and comprehension.

Reviewer #2 (Comments to the Authors (Required)):

While phagocytosis by RPE cells through the TAM receptor MerTK is undoubtedly an important process, its molecular underpinnings have not been explored in great detail so far. In this manuscript, Zihni et al. describe in great detail a molecular pathway downstream of MerTK that leads to the actin cytoskeleton reorganization involved in phagocytosis by RPEs. Using a wealth of quantitative experiments, they convincingly demonstrate the engagement of MerTK leads to the recruitment of the Cdc42 GEF Dbl3. This leads to the recruitment of the Cdc42 effectors N-WASP and MRCK β . All these molecules appeared to be important for POS internalization. Importantly, the authors demonstrated that this pathway plays an important role *in vivo*. They also went on to show the recruitment of focal adhesion proteins, which could provide a mechanical link to the actin cytoskeleton. Furthermore, they found that the role of MRCK β is conserved in macrophages. In general, the manuscript is clearly written and is built upon well-quantified experiments, despite a few missing controls. Thus, this study definitely represents a valuable contribution to the field. Where it falls short is in the lack of demonstration of the role of myosin contractility in this process. While non-muscle myosin II is a downstream effector of MRCK β and is recruited to the phagocytic cup, the role of myosin contractility in the internalization process, if any, remained far from being clear. As the authors repeatedly stressed the role of myosin contractility, including in the title, they need to demonstrate it or revise their manuscript accordingly, before its publication.

Major comments:

Several critical controls required to demonstrate the validity of the approaches are not presented in the current manuscript. For example, western blots showing the silencing of Dbl3 and MRCK β are missing. No validation of the CRISPR-Cas9 approach is presented either. Moreover, controls to demonstrate that the observed phenotypes are not due to off-target effects would also be useful.

The specificity of several tools is also questionable. Do the authors have any evidence that the "anti Cdc42-GTP" antibody recognizes only the active conformation of CDC42 in immunofluorescence experiments? The specificity of Wiskostatin has also been questioned in several studies. Is a similar phenotype observed with the N-WASP siRNA?

The authors are proposing that MRCK β is promoting POS internalization by activating myosin II. In the manuscript, they built this idea upon the notion that myosin II plays a role in particle internalization through Fc-mediated phagocytosis. However, this notion is controversial since blebbistatin, a specific inhibitor of Myosin II, reduced internalization in some studies but not others (Tsai et al. JCB 2008; Rotty et al. Dev Cell 2017), while older studies were based on poorly specific inhibitors (e.g. BDM). Thus, the manuscript should reflect the current state of the field. The authors also stated that myosin II is required for the completion of the cup closure. If the authors are aware of a research article demonstrating this function, they should cite it instead of a review. More importantly, the authors did not show whether inhibition of myosin II had an effect on POS internalization. Are the morphological changes observed at the phagocytic cup upon blebbistatin treatment associated with a change of POS internalization? Furthermore, is the reduced internalization observed upon silencing of MRCK β due to a lack of myosin II activation? To address this, the authors could compare the effect of MRCK β silencing when cells have been treated with blebbistatin. Next, the authors claim that myosin contractility is important. However, myosin II assembled in minifilaments (which requires MLC phosphorylation) has an actin bundling and a contractile activity. In several contexts, the role of myosin II is independent of its contractile activity (Choi, Vicente-Manzanares et al. Nat Cell Biol 2008; Ma, Kovacs et al. PNAS 2012). So what is the evidence that contractility is important for establishing the morphology of the phagocytic cup or the internalization of POS?

Minor comments:

Figure 2, in wiskostatin treated cells, the reduction of phospho-MLC recruitment could be simply due to the reduction of F-actin at the phagocytic cup. The authors should provide more convincing evidence that N-WASP is linked to myosin II activity or revise/clarify their interpretation.

Page 6, the statement that talin is recruited to integrins in response to tension is incorrect.

Reviewer #3 (Comments to the Authors (Required)):

The manuscript by Zihni et al investigates the mechanisms downstream of MerTK, the major phagocytic receptor responsible for

the phagocytosis of photoreceptor outer segments by the retinal pigment epithelium (RPE), that lead to the completion of phagocytosis. The steps that guide pseudopodia around phagocytic targets and the sealing of the phagosome in particular are not well-understood, especially in the biologically important context of the RPE: Daily removal of the distal portions of photoreceptors needs to be efficient, otherwise this leads to retinal degeneration. The work at least topically therefore stands to be of translational importance and to widely improve our understanding of phagocytosis. Indeed, the authors use a series of elegant systems and approaches, toggling between primary RPE cells, RPE cell lines, macrophages, and the retina to address the role of their signaling pathway in phagocytosis (I will note that the preparation of the manuscript was really well done and the experimental efforts in total are commendable). In sum, the authors conclude that MerTK leads to the phosphorylation and activation of Cdc42 by its GEF Dbl3. Active Cdc42 is then important for canonical N-WASP nucleation of branched actin polymerization; the authors confirm that the nucleation promoting factor is essential for phagocytosis by the RPE.

The major new findings posit the activation MRCK, a myosin regulatory kinase that activates myosin II, as a necessary node of the Cdc42-GTP pathway that guides pseudopodia and ultimately seals them around the phagocytic target. This is somewhat surprising since recently, two major papers from James Bear (Rotty et al., *Dev Cell*, 2017) and Clare Waterman (Jaumouille et al, *NCB*, 2019) have independently concluded that myosin II, as inhibited by Blebbistatin, has no discernable effect on Fc-mediated phagocytosis or complement-mediated phagocytosis by macrophages respectively. The role of myosin II in phagocytosis has always been controversial (see review by Barger, Gautier, and Krendel for example). While MerTK may execute very different signaling that leads to phagocytosis that is dependent on myosin II, Zihni et al also test the effect of inhibiting MRCK in macrophages during their phagocytosis of opsonized zymosan. Here, they find that MRCK/myosin-II are in fact essential. In Figure 5D, the authors demonstrate that there is a 50% reduction in phagocytosis of serum-opsonized zymosan when MRCK is silenced in macrophages. In the very same figure, the imaging of the zymosan shows complete polymerization of F-actin around the target (Figure 5F). In this regard, it becomes apparent that the authors need to quantitatively assess phagocytic efficiency throughout the paper using gold standard approaches for staining the phagocytic targets for exposed (outside) and internalized (inside) components. Phagocytic efficiency would then be equal to (the # or particles completely inside)/(the total bound). For POS, this could be achieved by staining the PS with annexin V before and after permeabilization. For the experiments with macrophages, this could be done by using anti-zymosan and/or anti-IgG antibodies before and after permeabilization.

This also becomes especially important in the case of the RPE since the work of David Williams has demonstrated a role for Myosins (Myo7A) and F-actin in the translocation of the phagosome from the apical to the basolateral region of the RPE. It therefore is paramount to have rigorous quantification of phagocytic efficiency; F-actin stains and z slices do not suffice to know if a particle has indeed been engulfed. The activation of Myosin-II by MRCK may also be essential for the apical domain of the RPE to reform post phagocytosis where zealous ruffling events are calmed only by its activation and restoration of polarity, similar to what the authors showed previously for aPKC and Par6.

The authors should also attempt to use either KO cell lines or their siRNA approach when looking at pFAK and integrin clustering (Figure 3M-O) since the BDP inhibitor acts on both MRCK and ROCK; Rho/ROCK is expected to have a plethora of effects on integrin activation.

For the Dbl3 active mutant, how does this localize to the nascent phagocytic cup in the absence of MerTK signaling? Does this lead to integrin-mediated phagocytosis without MerTK? It would be interesting to speculate on this. The work of Raymond Birge (*JCS*, 2005) and review by Kwon and Freeman (*Frontiers in Immunology*, 2020) certainly suggest interplay between the two pathways. If the major role of MerTK is to activate Cdc42 via Dbl3, this would need to be controlled in space and time so the active mutant must be localized by something else. Are phosphoinositides involved? Based on the imaging in Figure 7E, Dbl appears better targeted to the sealed phagosomes rather than nascent phagosomes.

Minor items:

- p3 vinculin is misspelled
- separate values from units (2h should be 2 h, etc)
- p4 "Knockdown of Cdc42 OR MRCKb"
- S Figure 7: xy plane is taken from the basolateral side in the control and apical side of the experimental condition. A projection with inside/outside staining will make things much clearer.
- Interpretation of the BDP5290 results as being equal to MRCK inhibition are overstatements, since this drug also inhibits ROCK, so this should be worded more carefully.

July 8, 2022

JCB #202012042 Revised

Dear Dr. Wang and Dr. Casadio

Thank you for inviting us to resubmit a revised version of our manuscript entitled "Spatiotemporal Control of Actomyosin Contractility by MRCK β Signaling Drives Phagocytosis".

We are sorry for the long time it took us to complete the revision. However, addressing the criticisms as completely as possible required many experiments, which, together with the long interruptions caused by the pandemic, took a large amount of time. Additionally, the first author was looking for a job/fellowship, which took a considerable amount of his time.

To our best knowledge, there have been no publications since our last submission that impact on the novelty of our paper.

We have addressed all key points that you outlined, including more evidence on the role of myosin II in phagocytosis using the RPE model system, through specific experiments suggested by Reviewers #2 and #3. We have also carried out additional phagocytic assays as suggested by Reviewer 3 to measure phagocytic efficiency to clarify any differences in mechanism between RPE and macrophages at the cell cortex and to confirm the results of key assays reported in the original manuscript. We also performed additional experiments suggested by Reviewer #1, addressing MRCK α 's regulatory function in actin polymerization during phagocytic cup formation and its relationship with N-WASP signaling. Additional control experiments suggested by Rev#2 have also been included.

We have listed all changes in the attached reply to the reviewers' comments. We have also reorganized the text and figures to adjust the manuscript to the JCB format. This has led to changes to all figures and throughout the manuscript.

We hope that you agree with us that the revision addresses the criticisms raised and are looking forward to hearing from you.

Best wishes

Ceniz Zihni, Maria Balda and Karl Matter

Response to Reviewers

We would like to thank the three reviewers for their detailed and constructive comments and hope they agree with us that the additional data and changes we made clarified the issues raised.

Reviewer #1 (Comments to the Authors (Required)):

In the study by Zihni et al., they mainly deployed the retinal pigment epithelium (RPE), supplemented with CRISPR-Cas9 modified mice, macrophages, and iPSC, to investigate the role of the MerTK signaling in the various stages of phagocytosis. Notably, they found that myosin-II activator MRCKb is required for regulation of actomyosin contractility to drive phagocytosis. In this process, receptor MerTK activates the Cdc42-GEF Dbp3 through Src-mediated phosphorylation. Then, active Cdc42 stimulates N-WASP and MRCKb. N-WASP promotes actin-based pseudopod-like protrusions, while MRCKb activates myosin-II to limit actin assembly and induces deformation of pseudopods into cups. These data suggested local regulatory feedback between N-WASP-assisted actin polymerization and MRCKb-stimulated myosin-II activation, actin disassembly, and contractility. Additionally, they proposed that MRCKb-driven contractility also induces the assembly of a mechanosensory bridge consisting of activated FAK, talin, and vinculin, which promotes integrin clustering to guide membrane wrapping POS. In addition to the RPE system, they also tested the mechanistic conservatism of MerTK signaling during FcR-mediated macrophage phagocytosis and animals retinal integrity in vivo. Finally, through using a phosphomimetic Dbp3 mutant, they demonstrated that Dbp3-signaling is sufficient to bypass MerTK to induce phagocytosis in iPSC-derived retinitis pigmentosa RPE cells, highlighting that the major role of MerTK receptor is to activate MRCKb-signaling.

The findings of this study provide a nice addition to understanding how the phagocytosis is spatiotemporally regulated. In particular, it elaborates and consolidates the functional significance of the Cdc42-signaling in phagocytosis, which has been proposed in the field. I have some criticisms, which could help clarify some confusion and help understand the functional model presented in this manuscript.

Major comments:

Page 3, Cdc42 has been involved in regulating the formation and stability of cell junctions, primarily through actin and vesicular trafficking. In addition to the model proposed, does the activation of Cdc42 promote the directional transport of integrin and thus assist clustering?

Indeed, Cdc42 participates in numerous cellular roles that are distinguished by its spatiotemporal regulation of GEFs and GAPs. Dbp3 is an apical activator of Cdc42 that functions at the apical margin and along the apical membrane after junctions have formed; there is no effect on cell junction in Dbp3 loss-of-function models (Zihni et al., 2014). Similarly, analysis of Integrin $\alpha v \beta 5$ here does not provide any evidence that Cdc42 or Dbp3 signaling are involved in directional transport of integrin as POS binding is not affected in any of our experiments after loss of Cdc42, Dbp3 or MRCK function (Fig.1d; Fig.S1d; Fig 4a,e,j,k; Fig.S3g-i). Similarly, it has been previously reported POS is able to bind efficiently to MerTK deficient cells (Ramsden et al., 2017), indicating that particle binding is not affected. Knockdown of the downstream effector of MerTK, Cdc42 does

also not affect POS binding to the RPE membrane (Fig.1d; Fig.S1d). We have tried to make this important point clearer along the Results section where the relevant data are described (Page 4, paragraph 2).

Is there a time point choice for rescue experiments in iPSC, such as different stages of phagocytosis?

Primary RPE were used to characterize the different stages of phagocytosis as a normal model. Since RPE differentiated from patient-derived iPSC is defective in phagocytosis, the model was suitable to study the rescue of phagocytosis as a gain of function model. These rescue experiments were performed after 3h with POS, an amount of time that ensures complete internalization in functional RPE, to determine whether the molecular repair produced RPE cells that behaved as competent phagocytes. This has been highlighted in page 9, paragraph 2.

Page 5, "These results support a role for MRCKb driven contractility as a regulator of actin dynamics during protrusion maturation into cups." The MLC phosphorylation level should be measured after the knockdown of MRCKb to reach this conclusion. Please adjust the language.

A knockdown of MRCK β and a blot for pMLC have been performed (Fig 3m), in addition to using a chemical inhibitor (Fig 3l). Immunofluorescence analysis of pMLC at cups was already performed in the original manuscript, enabled the determination of how pMLC appears specifically at phagocytic cups (now shown in Fig3g-i). These new data are now described on page 5 (paragraph 3).

Page 5, "MRCK activity is not required for pseudopod induction but for remodeling of protrusions to form cups, particle wrapping, and internalization." The formation mechanism of pseudopod should be carefully discussed and preliminarily tested, which is very important for refining this research's conceptual advance. Also, this can help clarify the critical regulatory modes in the phagocytosis.

We had included experiments demonstrating that pseudopod formation requires N-WASP, a Cdc42 effector that promotes actin polymerization, that cooperates with MRCK, regulating cup morphogenesis and molecular clutch assembly to promote movement of POS into the cell. With additional experiments new figures have been added and this information is included in Fig.1-Fig.6). It should be noted that new figure 2, demonstrates that POS contact with the RPE apical membrane at a very early time point (10min), stimulates integrin $\alpha v \beta 5$ accumulation at contact sites and leads to the accumulation of Dbl3 and active Cdc42 at adhesion sites, along with the recruitment of MRCK and N-WASP, leading to protrusion induction followed by cup morphogenesis and internalization. We have highlighted these early events in the discussion.

Page 6, Fig 2u. This model is very interesting, but there is insufficient evidence to sustain this model directly. For example, knockdown of MRCK and Myosin II, or overexpression of MRCK and Myosin II should be used in cells overexpressing N-WASP to detect whether the phenotype is enhanced or weakened.

As also pointed out by reviewer 2, the model needed revising since the illustration of myosin II contractility being dependent on polymerization was not a specific mechanism

identified in the study but an already established property whereby myosin motors require actin filaments to function. The key point in the present study that we wanted to highlight was that MRCK/myosin function regulates actin polymerization and remodelling during pseudopod transformation into cups. We have performed additional experiments also requested by Reviewer 2, demonstrating that both MRCK β kd or Myosin II inhibition display the same defects during phagocytosis and inhibiting both components at the same time does not result in a change in the defect, further supporting that they are functioning in the same pathway (Fig.3q-v; Fig4q-v). We also performed additional experiments suggested by Reviewer 2 as outlined below.

Fig 3, in addition to the inactivation of MRCKb, other methods could be used to enhance the actin depolymerization and see if it has the same effect, thus verifying this model.

To clarify, an inhibition of MRCK β does not enhance actin depolymerization, as highlighted in the previous point, but in fact results in uncontrolled actin polymerization (Fig.4a,b, f, j, l, q, r, u). Therefore, to further verify this model we confirmed that overgrowth of pseudopods is due to N-WASP since knockdown of N-WASP and MRCK β simultaneously inhibits excess actin polymerization (Fig.5j-n). Similarly, we show that MRCK β inhibition-induced pseudopod overgrowth is indeed due to actin polymerization by demonstrating that the actin polymerization inhibitor Latrunculin A also inhibits MRCK β knockdown-dependent pseudopod overgrowth (Fig.5j-n).

Fig 3, does the use of inhibitors reduce the recruitment of clutch molecules? Please show.

We have added panels in figure 6, demonstrating that both MRCK β and N-WASP inhibitors reduce the recruitment of the talin clutch molecule (Fig.6m; Fig.S3f).

Fig 3p, the data to support this model is insufficient. For example, it is recommended to display the colocalization of clutch molecules with MRCK and/or integrin at different stages, as well as the F-actin level changes at the same time scale.

We have added new panels in fig 6 (old Fig.3), demonstrating increases in both actin and pMLC from nascent protrusions into cups (fig.6b,c,e,f), similar to clutch proteins, and also an increase in colocalization of MRCK β with Talin at POS-induced pseudopods and during transformation into cups (Fig. 6k,l).

In Fig 5g left pathway panel, there is an arrow pointing from MRCK to N-WASP, is there any evidence supporting this claim?

We would like to thank the reviewer to pointing this out. In fact, the arrow should have been a block sign, indicting a regulatory function for MRCK on N-WASP. Since figures 5g and 8 both contained summaries of the overall mechanism of RPE phagocytosis, we have now removed this figure and only kept old figure 8, which is now new figure 10l.

Given the model proposed in Fig 5g, it is recommended that Fig 4 and Fig 5 be moved to the end, and Fig 6 and Fig 7 are moved after Fig 3 to keep the story smooth.

As mentioned above, figure fig.5g has been removed and the model is shown in figure 10l. Based on comments by all three reviewers, and the addition of several new main

figures, the panels have been rearranged throughout the manuscript.

Minor comments:

Fig 1, Page 4, there is a lack of explanation for the labels, and the Y-axis label of the statistical graph is covered. It is recommended to add an RPE diagram for explanation purposes.

The panels have now been rearranged so that overlaps are hopefully all removed. We are not sure what labels the reviewer is referring to but have generally tried to improve the explanation of the data shown in figure 1, and all labels are also indicated in the panels themselves.

Page 4, POS binding assay, please explain it briefly. Also, please explain how to assay Cdc42 activation.

POS binding was quantified by counting fluorescently labeled POS particles attached and internalized to the cells at the end of the experiment. Total bound POS was the sum of particles bound to the surface of RPE and internalized since internalized particles are also required to bind to the integrin α v β 5 receptor in the first instance (as shown by Finneman and colleagues). We have also added an additional experiments to further validate our assay at the request of Reviewer 3, (now clarified on page 4 (paragraph 2), page6 paragraph 1) and in more detail in the Materials and Methods, Statistical analysis).

We also clarified how we monitored Cdc42 activation (page 4, paragraph 2). This was done using a commercially available monoclonal antibody that specifically recognizes Cdc42 in its GTP-bound form. We have previously reported data supporting the specificity of this antibody, demonstrating that it only stains Cdc42 in immunofluorescence assays upon activation by Cdc42 exchange factors (Elbediwy et al., JCB 198, 677-693, 2012; the supplementary data in that paper show the results of stainings obtained with and without transfection of Cdc42 exchange factors).

Figure 1i, red arrows are hard to find.

The red arrows were in supplementary figure S1g, and have now been added to the main figure 1i.

The POS label in Fig 1j is unclear. Please indicate.

The position of the POS label is now indicated by white arrows. The label itself was not included to not obscure the analysis of F-actin. However, the POS label is included in the XY and Z-Sections in figure S1i, j.

It is recommended to give a schematic diagram of the early and late events, adding into Fig. 3, which helps readers understand the significance and importance of this part of data.

A new figure 2 has been added introducing the stages of membrane cup formation, i.e., nascent adhesions and protrusions that remodel into cups. Previous figure 3 has now been divided into new figures 4 and 6 due to the new experiments added, which together with the modified results section should make the results clearer.

Fig 3, EM shows that the cup cannot be observed after the BDP5290 application. Is this the reason for the lack of phagocytosis? The statistical information of c, f, h is incomplete, and no specific statistical indicators can be seen.

The lack of phagocytosis is due to a dysregulation of cup formation resulting in the inability of cups to wrap and engulf POS (Fig.3n-p; Fig.4a-o). The aim of the EM analysis was to serve as another method to measure phagocytosis. Each of those quantification panels contained means, standard deviations, individual data points, number of analysed cells, and p-values. We tried to clarify the appearance of those graphs in the revised version (now figure 4).

Fig 4, j matches l, but i and k do not match; please verify this. Also, please indicate where the h was cropped out. I did not see phagosomes in MRCK-deleted cells.

The confusion was due to an extra zoom image in the previous version (k). This has now been removed and the corresponding Zoom H and I and J and K have been highlighted. The reason why phagosomes were not observed in MRCK-deleted cells is due to the inhibitory effect of MRCK β knockout on phagocytosis.

Viculin is a typo, should be vinculin.

This has been corrected.

In general, the lack of annotations for many symbols in figure legends leads to difficulties in reading and comprehension.

We have tried to make the figure legends clearer for the readers.

Reviewer #2 (Comments to the Authors (Required)):

While phagocytosis by RPE cells through the TAM receptor MerTK is undoubtedly an important process, its molecular underpinnings have not been explored in great detail so far. In this manuscript, Zihni et al. describe in great detail a molecular pathway downstream of MerTK that leads to the actin cytoskeleton reorganization involved in phagocytosis by RPEs. Using a wealth of quantitative experiments, they convincingly demonstrate that the engagement of MerTK leads to the recruitment of the Cdc42 GEF Dbl3. This leads to the recruitment of the Cdc42 effectors N-WASP and MRCK β . All these molecules appeared to be important for POS internalization. Importantly, the authors demonstrated that this pathway plays an important role *in vivo*. They also went on to show the recruitment of focal adhesion proteins, which could provide a mechanical link to the actin cytoskeleton. Furthermore, they found that the role of MRCK β is conserved in macrophages. In general, the manuscript is clearly written and is built upon well-quantified experiments, despite a few missing controls. Thus, this study definitely represents a valuable contribution to the field. Where it falls short is in the lack of demonstration of the role of myosin contractility in this process. While non-muscle myosin II is a downstream effector of MRCK β and is recruited to the phagocytic cup, the role of myosin contractility in the internalization process, if any, remained far from being clear. As the authors repeatedly stressed the role of myosin contractility, including in the title, they need to demonstrate it or revise their manuscript accordingly, before its

publication.

Major comments:

Several critical controls required to demonstrate the validity of the approaches are not presented in the current manuscript. For example, western blots showing the silencing of Dbl3 and MRCK β are missing. No validation of the CRISPR-Cas9 approach is presented either. Moreover, controls to demonstrate that the observed phenotypes are not due to off-target effects would also be useful.

We have added additional panels in figures 4p, S1c and S3c, demonstrating knockdown of Dbl3 and MRCK β by western blotting. To confirm the effects of Dbl3 and MRCK β are not due to off target effects, we added data showing that two Dbl3 or MRCK β siRNA sequences targeting distinct regions of each gene, knock down each protein (S1c and S3c), and carried out phagocytosis efficiency assays using de-convoluted siRNAs that confirm that Dbl3 and MRCK β are required for phagocytosis of POS in RPE (Fig.S1a,b and Fig.S3a,b). As shown in different parts of the manuscript, a small molecule inhibitor of MRCK also phenocopies the MRCK β knockdown, further supporting the specificity of the loss of function phenotype. In vivo, we use immunofluorescence to demonstrate that CRISPR-Cas9-GFP subretinally injected into mice specifically transduces RPE cells specifically (FigS4a). We also show that MRCK β CRISPR knockout specifically occurs in the RPE using two distinct CRISPR viruses that target distinct regions of the MRCK β gene and that both disrupt apical F-actin distribution in the RPE in a comparable manner (Fig.7c,d;Fig.S4d-h).

The specificity of several tools is also questionable. Do the author have any evidence that the "anti Cdc42-GTP" antibody recognizes only the active conformation of CDC42 in immunofluorescence experiments? The specificity of Wiskostatin has also been questioned in several studies. Is a similar phenotype observed with the N-WASP siRNA?

We have previously published the specificity controls for the anti-Cdc42-GTP monoclonal antibody in immunofluorescence assays. By transfecting different Cdc42 guanine nucleotide exchange factors, we demonstrated induction of Cdc42 activity induces staining by the Cdc42-GTP antibody (Supplementary data, Elbediwy et al., JCB 198, 677-693, 2012). We also performed knockdown experiments demonstrating that Cdc42-GTP staining by this monoclonal antibody in epithelial cells is indeed Cdc42-dependent (Supplementary data, Elbediwy et al., JCB 198, 677-693, 2012). We also performed parallel experiments with this monoclonal anti-Cdc42-GTP antibody and a Cdc42 FRET biosensor, demonstrating that both probes for active Cdc42 result in comparable effects on the cellular Cdc42-GTP distribution upon knockdown of a junctional Cdc42 GAP (Elbediwy et al., JCB 198, 677-693, 2012) or inactivation of apical Cdc42 activation by depletion of Dbl3 (Zihni et al., 2014). We have also added a new panel demonstrating that the phenotype observed with the inhibitor Wiskostatin is the same as the one induced by the N-WASP siRNA (Fig.5f-i).

The authors are proposing that MRCK β is promoting POS internalization by activating myosin II. In the manuscript, they built this idea upon the notion that myosin II plays a role in particle internalization through Fc-mediated phagocytosis. However, this notion is controversial since blebbistatin, a specific inhibitor of Myosin II, reduced internalization

in some studies but not others (Tsai et al. JCB 2008; Rotty et al. Dev Cell 2017), while older studies were based on poorly specific inhibitors (e.g. BDM). Thus, the manuscript should reflect the current state of the field.

We have modified the text throughout the manuscript to reflect the current state of the field, indicating that the role of myosin II in internalization of particles by macrophages, is controversial.

The authors also stated that myosin II is required for the completion of the cup closure. If the authors are aware of a research article demonstrating this function, they should cite it instead of a review.

This text in the discussion has been corrected to make a distinction of a possible role for MRCK β after cup closure rather than during based on functions proposed for myosin-II, including specific references (Ghandi et al.,2009; Jaumouille and Waterman, 2020) in page 10/11).

More importantly, the authors did not show whether inhibition of myosin II had an effect on POS internalization.

Are the morphological changes observed at the phagocytic cup upon blebbistatin treatment associated with a change of POS internalization? Furthermore, is the reduced internalization observed upon silencing of MRCK β due to a lack of myosin II activation? To address this, the authors could compare the effect of MRCK β silencing when cells have been treated with blebbistatin.

In figure 3n-p, we showed that myosin II is important for normal cup morphology and centering POS. Therefore, we used a time point of 30 min, the optimal time for cup assembly, to study this process. We have now added new panels in figure 3q-v after 3h (1h particle binding and 2h chase) to study internalization, at a time point when most cups have disassembled after POS internalization. We demonstrate that blebbistatin inhibition of myosin II results in actin-rich protrusion overgrowth, particles stuck around this disorganized structure, and inhibition of POS internalization. We further confirm that phagocytic efficiency is inhibited by blebbistatin after this long timepoint (Fig.S2j,k). When we compare actin rich disorganized protrusion formation via inhibition of phagocytosis by blebbistatin or MRCK β kd, both are similar, and inhibiting myosin-II in MRCK β kd cells, as the reviewer proposed, resulted in a similar phenotype, indicating that MRCK β and myosin-II indeed function in the same pathway (Fig.4.q-v).

Next, the authors claim that myosin contractility is important. However, myosin II assembled in minifilaments (which requires MLC phosphorylation) has an actin bundling and a contractile activity. In several contexts, the role of myosin II is independent of its contractile activity (Choi, Vicente-Manzanares et al. Nat Cell Biol 2008; Ma, Kovacs et al. PNAS 2012). So what is the evidence that contractility is important for establishing the morphology of the phagocytic cup or the internalization of POS?

Here, we have shown that inhibition of MLC phosphorylation by MRCK inactivation inhibits cup morphogenesis and POS internalization. As the reviewer suggested, we performed experiments demonstrating that MRCK and myosin-II indeed function in the same pathway (see previous comment). We have also shown inhibition of phagocytosis with the myosin-II specific inhibitor blebbistatin (Kovacs et al., 2004).

Blebbistatin does not inhibit actin binding but its motor activity, supporting the conclusion that it is indeed contractility that is important. Choi et al. used blebbistatin to demonstrate the myosin function they studied was motor-independent (i.e., not inhibited by blebbistatin). However, Kovacs et al. suggest that Blebbistatin may also generate tension and cytokinetic ring contraction in a motor-independent manner. Hence, we have now added these references to the text and discussed the different possibilities for the role of myosin-II in phagocytosis in relation of MRCK signaling. However, despite being motor independent, the consequence of myosin-II function was still contraction of the cytokinetic ring (i.e., actomyosin contraction).

Minor comments:

Figure 2, in wiskostatin treated cells, the reduction of phospho-MLC recruitment could be simply due to the reduction of F-actin at the phagocytic cup. The authors should provide more convincing evidence that N-WASP is linked to myosin II activity or revise/clarify their interpretation.

We thank the reviewer for pointing this out. We in fact did mean to indicate that the reduction in pMLC-accumulation at the phagocytic cup was due to a reduction of F-actin. The sentence has been revised to be make it clearer.

Page 6, the statement that talin is recruited to integrins in response to tension is incorrect.

We have corrected this error (now page 6, last paragraph, of the corrected manuscript).

Reviewer #3 (Comments to the Authors (Required)):

The manuscript by Zihni et al investigates the mechanisms downstream of MerTK, the major phagocytic receptor responsible for the phagocytosis of photoreceptor outer segments by the retinal pigment epithelium (RPE), that lead to the completion of phagocytosis. The steps that guide pseudopodia around phagocytic targets and the sealing of the phagosome in particular are not well-understood, especially in the biologically important context of the RPE: Daily removal of the distal portions of photoreceptors needs to be efficient, otherwise this leads to retinal degeneration. The work at least topically therefore stands to be of translational importance and to widely improve our understanding of phagocytosis. Indeed, the authors use a series of elegant systems and approaches, toggling between primary RPE cells, RPE cell lines, macrophages, and the retina to address the role of their signaling pathway in phagocytosis (I will note that the preparation of the manuscript was really well done and the experimental efforts in total are commendable). In sum, the authors conclude that MerTK leads to the phosphorylation and activation of Cdc42 by its GEF Dbl3. Active Cdc42 is then important for canonical N-WASP nucleation of branched actin polymerization; the authors confirm that the nucleation promoting factor is essential for phagocytosis by the RPE.

The major new findings posit the activation MRCK β , a myosin regulatory kinase that activates myosin II, as a necessary node of the Cdc42-GTP pathway that guides

pseudopodia and ultimately seals them around the phagocytic target. This is somewhat surprising since recently, two major papers from James Bear (Rotty et al., Dev Cell, 2017) and Clare Waterman (Jaumouille et al, NCB, 2019) have independently concluded that myosin II, as inhibited by Blebbistatin, has no discernable effect on Fc-mediated phagocytosis or complement-mediated phagocytosis by macrophages respectively. The role of myosin II in phagocytosis has always been controversial (see review by Barger, Gautier, and Krendel for example).

While MerTK may execute very different signaling that leads to phagocytosis that is dependent on myosin II, Zihni et al also test the effect of inhibiting MRCK in macrophages during their phagocytosis of opsonized zymosan. Here, they find that MRCK/myosin-II are in fact essential. In Figure 5D, the authors demonstrate that there is a 50% reduction in phagocytosis of serum-opsonized zymosan when MRCK is silenced in macrophages. In the very same figure, the imaging of the zymosan shows complete polymerization of F-actin around the target (Figure 5F). In this regard, it becomes apparent that the authors need to quantitatively assess phagocytic efficiency throughout the paper using gold standard approaches for staining the phagocytic targets for exposed (outside) and internalized (inside) components. Phagocytic efficiency would then be equal to (the # or particles completely inside)/(the total bound). For POS, this could be achieved by staining the PS with annexin V before and after permeabilization. For the experiments with macrophages, this could be done by using anti-zymosan and/or anti-IgG antibodies before and after permeabilization.

The reviewer raises an important point, and we now added data on phagocytic efficiency in both RPE cells and macrophages. To measure phagocytic efficiency, we used approaches described for RPE (Mazzoni et al., 2019) and also for macrophages (Jaumouille et al., 2019), when the particle is already labelled, which gives a more robust reading. In this method, cells are fixed without permeabilization. Consequently, total labelled particles represent total bound POS, and particles also stained by an antibody represent external bound particle labelling. The number of internalized particles is then divided by the number of total bound particles to give a value for phagocytic efficiency. The use of Z-sections of F-actin-stained cells was important to directly study any correlation between internalization and apical membrane morphogenesis in RPE cells. New panels analyzing phagocytic efficiency (Fig.S1a,b; Fig.S2j,k; Fig.S3a,b) further verify our results, demonstrating that Dbl3, MRCK β and myosin-II are required for POS internalization. In macrophages, we tested whether MRCK β possessed a conserved role in phagocytic membrane remodeling and found that there was an accumulation of Zymosan at the cortex. As the reviewer correctly pointed out, it was not clear whether the particles accumulating at the cortex were internalized or not. Analysis of phagocytic efficiency indicates that particles do internalize as efficiently as controls (Fig.9e,f). However, the phagocytic vacuoles stay at the membrane cortex. Therefore, our results indicate a role for MRCK in cortical remodeling during macrophage phagocytosis, at a later stage than during RPE phagocytosis. This further supports our conclusion that MRCK plays a conserved yet diverse role at the cortex during phagocytosis. We have added a paragraph in the discussion, considering these new data (page 11, paragraph1).

This also becomes especially important in the case of the RPE since the work of David Williams has demonstrated a role for Myosins (Myo7A) and F-actin in the translocation of the phagosome from the apical to the basolateral region of the RPE. It therefore is paramount to have rigorous quantification of phagocytic efficiency; F-actin stains and z slices do not suffice to know if a particle has indeed been engulfed.

As indicated above, assessments of phagocytic efficiency in RPE show that it is strongly reduced when MRCK β or its substrate myosin-II are inhibited, supporting the conclusion that both are required before particles are internalized by controlling remodeling of nascent protrusions into cups to internalize POS using a molecular clutch mechanism. The work of David Williams suggests that internalized phagosomes are passed onto myosin7A that in turn passes phagosomes onto apical minus ends of polarized microtubules; phagosomes are then transported towards the basolateral regions of the RPE for subsequent degradation. Therefore, our results suggest that myosin-II functions upstream of myosin 7a.

The activation of Myosin-II by MRCK may also be essential for the apical domain of the RPE to reform post phagocytosis where zealous ruffling events are calmed only by its activation and restoration of polarity, similar to what the authors showed previously for aPKC and Par6.

As the reviewer suggests, the activation of myosin-II by MRCK β may also be required for reformation of the apical domain post engulfment, in addition to regulating cup formation and remodelling, and dynamics. However, our data here show that taking cells that are fully polarized and have not seen any POS since plating require the MRCK pathway for phagocytosis (e.g., short term inhibition with small molecules blocks phagocytosis but not POS binding). However, it will certainly be interesting to test whether the Dbl3/MRCK also plays a role in recovering a fully functional apical membrane after a round of phagocytosis.

The authors should also attempt to use either KO cell lines or their siRNA approach when looking at pFAK and integrin clustering (Figure 3M-O) since the BDP inhibitor acts on both MRCK and ROCK; Rho/ROCK is expected to have a plethora of effects on integrin activation.

We have added new panels in figure 6 (Fig.6r,s), demonstrating that siRNA mediated kd of MRCK β in primary RPE cells results in the same phenotypic effects as the use of BDP inhibitor on integrin clustering. Knockdown of MRCK in primary RPE also results in an attenuation of pFAK, similar to the use of the MRCK inhibitor (Fig.6o).

For the Dbl3 active mutant, how does this localize to the nascent phagocytic cup in the absence of MerTK signaling? Does this lead to integrin-mediated phagocytosis without MerTK? It would be interesting to speculate on this. The work of Raymond Birge (JCS, 2005) and review by Kwon and Freeman (Frontiers in Immunology, 2020) certainly suggest interplay between the two pathways. If the major role of MerTK is to activate Cdc42 via Dbl3, this would need to be controlled in space and time so the active mutant must be localized by something else. Are phosphoinositides involved? Based on the

imaging in Figure 7E, Dbl appears better targeted to the sealed phagosomes rather than nascent phagosomes.

Since integrin is required to bind POS and control the activity of MerTK signaling, activating the MerTK effector pathway via Dbl3 without functional MerTK would still require the Integrin receptor. Integrin receptor expression and localization is normal in MerTK deficient RPE cells (Ramsden et al., 2017). We found in this study that Dbl3 localizes to nascent adhesions at POS-integrin binding sites, even in the absence of MerTK agreeing with the previous study that suggests integrin signaling is not affected by MerTK signaling. Previous studies including from our lab suggest that apical localization of Dbl3 is due to more than one interaction. Dbl3 interacts with ezrin via its PH domain, which may be affected by phosphoinositides. Ezrin is required for apical recruitment of Dbl3. However, ezrin binding is not sufficient as a second domain, the CRAL-TRIO/Sec14 domain is also required. The CRAL-TRIO domain is only intact in Dbl3 and not other Dbl isoforms; hence, the specificity of Dbl3 for apical signaling mechanisms (Zihni et al., 2014). CRAL-TRIO/Sec14 domains are also thought to bind lipids but their specificity is not well defined. We have added a sentence in the discussion, page 11, paragraph 3 lines 7-9.

Figure 7e (now Fig. 10) represents a long time point of 3h, highlighted in page 10, with POS that is, in normal cells, sufficient for complete internalization as the aim of the experiments was to determine whether Dbl3 rescues phagocytosis deficiency due to MerTK impairment. However, we demonstrate earlier in a time course analysis using wild type RPE that Dbl3 indeed localizes to sites of POS-integrin $\alpha v \beta 5$ adhesion at the onset of nascent protrusion induction (Fig.1k). Therefore, whilst Dbl3 localizes to phagosomes, its initially recruited to POS adhesion sites where it activates Cdc42 and its effectors MRCK β and N-WASP.

Minor items:

-p3 vinculin is misspelled

This has been corrected

-separate values from units (2h should be 2 h, etc)

This has been corrected

-p4 "Knockdown of Cdc42 OR MRCKb"

This has been corrected

-S Figure 7: xy plane is taken from the basolateral side in the control and apical side of the experimental condition. A projection with inside/outside staining will make things much clearer.

The purpose of these images was to display the increased actin structures that occur in MRCK β deficient cell membranes, which is best analysed by XY analysis. Whilst part of the area examined in the control panels was indeed more basolateral due to an uneven layer, we have now included complete apical control sections comparable to experimental conditions (now shown in Figure S3d,e).

-Interpretation of the BDP5290 results as being equal to MRCK inhibition are overstatements, since this drug also inhibits ROCK, so this should be worded more carefully.

We have highlighted in the manuscript that the BDP5290 inhibitor displays more specificity towards MRCK β than ROCK as demonstrated by studies from Michael Olson's Lab (Unbekandt et al., 2014), page 5 Experiments were complemented with new MRCK β knockdown experiments that resulted in similar defects in POS-activated MLC phosphorylation, phagocytosis and membrane cup remodeling as shown in Figure 3m; Figure 4q; Figure 6o,r; as well as previously displayed data using primary RPE, Figure 4j, and in vivo data in figure 9.

Additional references

Mazzoni, F., Y. Mao, and S.C. Finnemann. 2019. Advanced Analysis of Photoreceptor Outer Segment Phagocytosis by RPE Cells in Culture. *Methods Mol Biol.* 1834:95-108.

July 25, 2022

RE: JCB Manuscript #202012042R

Prof. Karl Matter
University College London
Division of Cell Biology UCL Institute of Ophthalmology University College London
Bath Street
London EC1V 9EL
United Kingdom

Dear Prof. Matter:

Thank you for submitting your revised manuscript entitled "Spatiotemporal Control of Actomyosin Contractility by MRCKbeta Signaling Drives Phagocytosis". We would be happy to publish your paper in JCB pending final revisions necessary to meet our formatting guidelines (see details below).

A. MANUSCRIPT ORGANIZATION AND FORMATTING:

- 1) Text limits: Character count for Articles is < 40,000, not including spaces. Count includes abstract, introduction, results, discussion, and acknowledgments. Count does not include title page, figure legends, materials and methods, references, tables, or supplemental legends.
- 2) Figures limits: Articles may have up to 10 main text figures.
- 3) Figure formatting: Scale bars must be present on all microscopy images, including inset magnifications. Molecular weight or nucleic acid size markers must be included on all gel electrophoresis.
- 4) Statistical analysis: Error bars on graphic representations of numerical data must be clearly described in the figure legend. The number of independent data points (n) represented in a graph must be indicated in the legend. Statistical methods should be explained in full in the materials and methods. For figures presenting pooled data the statistical measure should be defined in the figure legends. Please also be sure to indicate the statistical tests used in each of your experiments (either in the figure legend itself or in a separate methods section) as well as the parameters of the test (for example, if you ran a t-test, please indicate if it was one- or two-sided, etc.). Also, if you used parametric tests, please indicate if the data distribution was tested for normality (and if so, how). If not, you must state something to the effect that "Data distribution was assumed to be normal but this was not formally tested."
- 5) Abstract and title: The abstract should be no longer than 160 words and should communicate the significance of the paper for a general audience. The title should be less than 100 characters including spaces. Make the title concise but accessible to a general readership.
- 6) Materials and methods: Should be comprehensive and not simply reference a previous publication for details on how an experiment was performed. Please provide full descriptions in the text for readers who may not have access to referenced manuscripts.
- 7) Please be sure to provide the sequences for all of your primers/oligos and RNAi constructs in the materials and methods. You must also indicate in the methods the source, species, and catalog numbers (where appropriate) for all of your antibodies. Please also indicate the acquisition and quantification methods for immunoblotting/western blots.
- 8) Microscope image acquisition: The following information must be provided about the acquisition and processing of images:
 - a. Make and model of microscope
 - b. Type, magnification, and numerical aperture of the objective lenses
 - c. Temperature
 - d. Imaging medium
 - e. Fluorochromes
 - f. Camera make and model

g. Acquisition software

h. Any software used for image processing subsequent to data acquisition. Please include details and types of operations involved (e.g., type of deconvolution, 3D reconstitutions, surface or volume rendering, gamma adjustments, etc.).

10) Supplemental materials: There are strict limits on the allowable amount of supplemental data. Articles may have up to 5 supplemental figures. Please also note that tables, like figures, should be provided as individual, editable files. A summary of all supplemental material should appear at the end of the Materials and methods section.

13) ORCID IDs: ORCID IDs are unique identifiers allowing researchers to create a record of their various scholarly contributions in a single place. At resubmission of your final files, please consider providing an ORCID ID for as many contributing authors as possible.

Please note that JCB now requires authors to submit Source Data used to generate figures containing gels and Western blots with all revised manuscripts. This Source Data consists of fully uncropped and unprocessed images for each gel/blot displayed in the main and supplemental figures. Since your paper includes cropped gel and/or blot images, please be sure to provide one Source Data file for each figure that contains gels and/or blots along with your revised manuscript files. File names for Source Data figures should be alphanumeric without any spaces or special characters (i.e., SourceDataF#, where F# refers to the associated main figure number or SourceDataFS# for those associated with Supplementary figures). The lanes of the gels/blots should be labeled as they are in the associated figure, the place where cropping was applied should be marked (with a box), and molecular weight/size standards should be labeled wherever possible.

B. FINAL FILES:

**The license to publish form must be signed before your manuscript can be sent to production. A link to the electronic license to publish form will be sent to the corresponding author only. Please take a moment to check your funder requirements before

choosing the appropriate license.**

Thank you for this interesting contribution, we look forward to publishing your paper in Journal of Cell Biology.

Sincerely,

Xiaochen Wang, PhD
Monitoring Editor

Andrea L. Marat, PhD
Senior Scientific Editor

Journal of Cell Biology

Reviewer #1 (Comments to the Authors (Required)):

As I stated before, the findings of this study provide an excellent addition to understanding how phagocytosis is spatiotemporally regulated. It elaborates and consolidates the functional significance of the Cdc42-signaling in phagocytosis. Furthermore, in the past two years, the authors rationally sorted out all concerns and gave plausible explanations one by one, either through experiments or literature analysis. Given the current data quality and underlying scientific logic, this study already meets the JCB's innovative requirements, and I support its publication.

Reviewer #3 (Comments to the Authors (Required)):

The manuscript by Zihni et al. very nicely and thoroughly establishes a novel role for actomyosin contractility in phagocytosis by the retinal pigment epithelium. The work identifies new roles for MRCKB, Cdc42/Dbp3, and firmly demonstrates molecular differences in phagocytosis by the RPE from that of macrophages.

I commend the effort in addressing the reviewer comments including my own. At this stage, I hope the other reviewers find the work to be as sufficiently meritorious of publication in JCB as I do.